# Understanding Arrhythmogenic Cardiomyopathy: Advances through the Use of Human Pluripotent Stem Cell Models

**DOI:** 10.3390/genes14101864

**Published:** 2023-09-25

**Authors:** Christianne J. Chua, Justin Morrissette-McAlmon, Leslie Tung, Kenneth R. Boheler

**Affiliations:** 1Department of Biomedical Engineering, Johns Hopkins University School of Medicine, Baltimore, MD 21205, USA; cchua3@jh.edu (C.J.C.); jmorri65@jh.edu (J.M.-M.); ltung@jh.edu (L.T.); 2Division of Cardiology, Department of Medicine, Johns Hopkins University School of Medicine, Baltimore, MD 21205, USA

**Keywords:** arrhythmogenic cardiomyopathy, heart, desmosome, human induced pluripotent stem cells, cardiomyocytes, arrhythmia

## Abstract

Cardiomyopathies (CMPs) represent a significant healthcare burden and are a major cause of heart failure leading to premature death. Several CMPs are now recognized to have a strong genetic basis, including arrhythmogenic cardiomyopathy (ACM), which predisposes patients to arrhythmic episodes. Variants in one of the five genes (*PKP2, JUP, DSC2, DSG2,* and *DSP*) encoding proteins of the desmosome are known to cause a subset of ACM, which we classify as desmosome-related ACM (dACM). Phenotypically, this disease may lead to sudden cardiac death in young athletes and, during late stages, is often accompanied by myocardial fibrofatty infiltrates. While the pathogenicity of the desmosome genes has been well established through animal studies and limited supplies of primary human cells, these systems have drawbacks that limit their utility and relevance to understanding human disease. Human induced pluripotent stem cells (hiPSCs) have emerged as a powerful tool for modeling ACM in vitro that can overcome these challenges, as they represent a reproducible and scalable source of cardiomyocytes (CMs) that recapitulate patient phenotypes. In this review, we provide an overview of dACM, summarize findings in other model systems linking desmosome proteins with this disease, and provide an up-to-date summary of the work that has been conducted in hiPSC-cardiomyocyte (hiPSC-CM) models of dACM. In the context of the hiPSC-CM model system, we highlight novel findings that have contributed to our understanding of disease and enumerate the limitations, prospects, and directions for research to consider towards future progress.

## 1. Introduction—Cardiomyopathies and ACM

Cardiomyopathies (CMPs) represent a group of acquired or hereditary syndromes that compromise cardiac function and often lead to premature death. Although the types (Figure 1) and causes (idiopathic/undefined versus familial/genetic) are diverse, CMPs often affect cell properties (sarcomeres, mitochondria, and metabolism), which contribute to abnormalities in myocardial structure and function (chamber dimensions, wall thickness, contractility, and stiffness), and electrical conduction (heart rhythms) [1]. The consequences of these defects often involve a reduction in cardiac output and, in some instances, sudden cardiac death (SCD) in young adults. All CMPs are associated with health issues linked with high rates of morbidity, mortality, and the need for clinical care. Clinical presentations of these syndromes are heterogeneous and range from individuals who are largely asymptomatic, to those with moderate to severe dysfunction, to those having exercise intolerance, and most egregiously, to those experiencing heart failure, arrhythmias, and death. 

Arrhythmogenic cardiomyopathy (ACM) is one of the leading causes of CMP-associated ventricular arrhythmias. To date, numerous genes have been identified that may cause or contribute to ACM (Figure 1); consequently, this syndrome encompasses a wide range of phenotypes that have little in common on an etiological and mechanistic basis. It is also a relatively rare disease affecting only 1 in 2000 to 1 in 5000 people among which ~60% of ACM patients are familial in origin. Over the past two decades, research on ACM has focused predominantly on the role of desmosomal gene mutations in adult heart. Desmosomes are prominent cell–cell adhesion sites between CMs that support the physical stability and integrity of heart tissues and are central to the formation of intercalated discs (ICD). Currently, it is believed that ~30–50% of all ACM cases are linked with genes that encode proteins of the desmosomes [2]. Desmosome-related arrhythmogenic cardiomyopathy (dACM) disproportionately affects males more than females [3,4]. The genetic basis of this disease is characterized, generally, by autosomal dominant inheritance but with a relatively low penetrance, as some individuals predisposed to dACM due to desmosomal mutations do not develop disease phenotypes. Some forms of dACM are autosomal recessive [5,6]. It is believed that ancillary genetic factors, genetic backgrounds, affected regions of the heart, as well as other non-genetic (age, exercise, and stress) factors affect its penetrance; however, the basis for this low penetrance is poorly understood [2]. The mechanisms of disease development thus remain somewhat ambiguous; consequently, management protocols are less clear than needed to treat patients with this syndrome. 

**Figure 1 genes-14-01864-f001:**
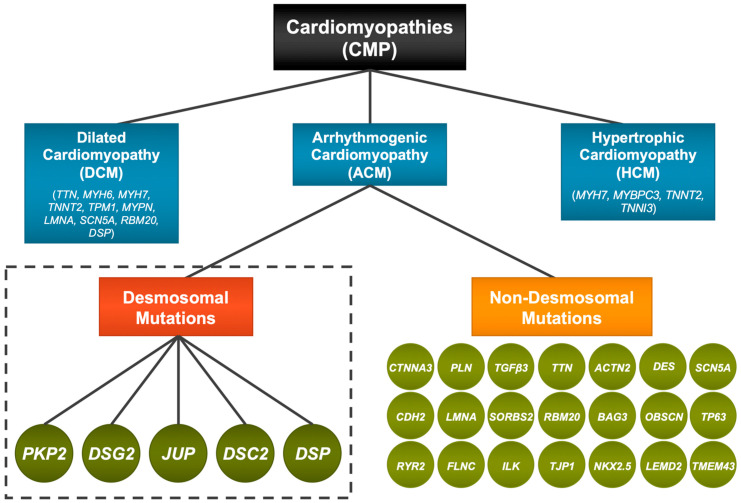
Gene mutations implicated in cardiomyopathies. Cardiomyopathies (CMPs) were conventionally classified according to the major clinical phenotypes manifested by the patient (see text); however, genetics has identified gene mutations that contributed to the manifestation of dilated cardiomyopathy (DCM), hypertrophic cardiomyopathy (HCM), and arrhythmogenic cardiomyopathy (ACM). Approximately half of patients with ACM have an identifiable genetic cause for disease. A majority of these mutations can be attributed to variants in five genes encoding proteins of the desmosome (red): *PKP2, DSG2, JUP, DSC2,* and *DSP* (dark green circles). Mutations in non-desmosomal genes (orange) also contribute to forms of ACM with distinct etiologies. First Row: CTNNA3 [7], PLN [8], TGFβ3 [9], TTN [10], ACTN2 [11], DES [12], SCN5A [13]; Second Row: CDH2 [14], LMNA [15], SORBS2 [16], RBM20 [17], BAG3 [18], OBSCN [19], TP63 [20]; Third Row: RYR2 [21], FLNC [22], ILK [23], TJP1 [24], NKX2..5 [24], LEMD2 [25], and TMEM43 [26].

Initially, most advances in our understanding of gene mutations responsible for dACM have come from human epidemiological and patient studies as well as functional analyses of transgenic mouse models and primary human cells. dACM is characterized by electrical (arrhythmias), structural (fibrofatty infiltration), and age-associated phenotypic abnormalities. However, significant differences in cardiac physiology between rodents and humans as well as limited availability of CMs from patients with dACM have hampered research efforts designed to better understand the mechanisms of these diseases [27]. Moreover, mouse models of ACM remain unproven for finding clinically feasible therapies [28]. As an alternative source of cells, CMs generated from human pluripotent stem cells (hPSCs, including both embryonic stem cells, ESC, and induced pluripotent stem cells, iPSCs) represent a viable and reproducible cell source for disease modeling, particularly for those models where a genetic cause can be attributed to the disease phenotype [29]. In addition, strategies to enrich lineage- or chamber-specific CMs have advanced the use of these cells to better understand how gene mutations may contribute to disease processes [30]. In vitro derived CMs that recapitulate key aspects of CMPs for applications in functional testing, drug screening, disease modelling, and physiologically relevant, bioengineered tissue models have the potential to transcend current experimental and therapeutic limitations. 

In writing this review, our goal is to provide a broad perspective on the genetic causes that directly link genes of the desmosome with ACM (Figure 1). An extensive overview of the clinical correlates of dACM or the management of this disease is not provided, as these areas have been reviewed extensively elsewhere [31,32]. Instead, we will focus on how mutations may contribute to compromised desmosome structures and to dACM. We will then discuss how human iPSC (hiPSC) models have been employed to unveil the molecular mechanisms and pathogenesis of dACM, as well as limitations and prospects for the use of these model systems. 

## 2. Clinical Phenotype

dACM was originally defined as a right ventricular CMP characterized by fibrofatty infiltration in the right ventricle and clinically characterized by arrhythmias, heart failure, syncope, and SCD [33]. Since this early description, dACM has been shown to involve not just the right ventricle, but also the left ventricle or both ventricles with regional or global cardiac dysfunction. The most affected anatomical regions usually involve the right ventricular inflow and outflow tracts and the right ventricular apex. However, when the left side of the heart is affected, the disease usually involves the subepicardial or midmyocardial layers of the free wall. Arrhythmias occur independently of ventricular dysfunction, consistent with aberrant electrical activity on a single- or multi-cellular level, even in the absence of obvious anatomical defects [31,32]. The development of dACM has four distinct phases that most frequently include (1) an early concealed phase when structural abnormalities are undetectable, but when electrical abnormalities may begin; (2) an overt electrical disorder characterized by ECG abnormalities that may include inverted T-waves and arrhythmias with left bundle branch block ventricular tachycardia; (3) a typically overt right ventricular disease with a loss of healthy myocardium with fibrofatty replacement that leads to global right ventricular dysfunction; and (4) an end-stage heart failure with biventricular involvement [34]. Histologically, this syndrome is characterized by distinct patterns of fibrosis when compared with other CMPs [35]. Cardiomyocyte hypertrophy is not always observed [36].

Clinically, the diagnosis of dACM is challenging and involves the use of diverse criteria from multiple testing categories (arrhythmias, depolarization abnormalities, repolarization abnormalities, anatomical defects detected via imaging, family history/genetics, and histology). Criteria for clinical diagnoses can be found in the recommendations of the 2010 consensus International Task Force (ITFC) [37]. The recognition of left and biventricular forms of dACM have led to additional criteria for the diagnosis of ACM, including morpho-functional and tissue abnormalities, structural myocardial, repolarization, and depolarization abnormalities of the left ventricle, as well as left ventricular arrhythmias [38]. 

Currently, about 30–50% of ACM cases can be attributed to a disease-causing mutation in a desmosome-encoding gene [39]. These mutations have been observed in plakophilin-2 (*PKP2*) (the most frequently involved), plakoglobin (*JUP*), desmoplakin (*DSP*), desmoglein-2 (*DGS2*), and desmocollin-2 (*DSC2*). Multiple variants exist for each of these genes. Non-desmosomal genes also have been associated with ACM with either definitive or moderate evidence [40]. Examples include transmembrane protein 43 (TMEM43), desmin (DES), phospholamban (PLN), titin (TTN), and N-cadherin (CDH2) among others (Figure 1). Individuals may display compound heterozygosity with two separate variants occurring in the same gene or digenic mutations where gene variants occur in more than one gene. Although usually autosomal dominant, reduced penetrance with non-autosomal dominance has been observed. Examples include a homozygous mutation in *JUP* linked to Naxos disease, which is common to some individuals in Greece and is characterized by onset of symptoms during childhood that include both cardiac and cutaneous involvement. A second autosomal recessive form, also a cardiocutaneous disorder, is called Carvajal-Huerta syndrome, which is caused by some *DSP* mutations [5,6]. 

## 3. Desmosomal Proteins and Structures

Normal myocardium requires that individual CMs align themselves to maximize force generation during systole [41]. Central to the maintenance of cell and tissue integrity during contraction are the intercalated discs (ICDs), which in adults localize at the longitudinal ends of the cardiomyocytes [42]. These structures are composed of desmosomes, adherens junctions (AJs, although also known as fascia adherens), gap junctions, and ion channels. Structurally, desmosomes and AJs constitute two independent junctional structures that link cells together mechanically through binding with the cell’s cytoskeleton. These junctions are physically distinguishable as plaques in epithelium, but in cardiomyocytes, only desmosome plaques are present [43]. Instead, AJs exist in a region known as the area composita, where AJs, gap junctions, and components of desmosomes exist as hybrid, mixed-type networks linked intracellularly through common adaptor proteins such as PKP, JUP, and catenin proteins [44,45,46]. Although these complexes were originally thought to exist solely as physically independent structures, the current model of the ICD reflects an understanding of how protein interactions within the “connexosome” result in colocalization of the entities [44]. 

Functionally, desmosomes provide strong adhesion between cells [47] ((1) in Figure 2A). The intra- and inter-cellular components of the desmosome link with intracellular intermediate filaments (IFs) of the cytoskeleton to form a network of adhesive bonds that provides mechanical strength to tissues. When desmosome adhesion fails, tissues subjected to mechanical forces may lose cohesion (Figure 2A2). As described above, components of the desmosome can be categorized into three protein families: cadherins, armadillo proteins, and plakins. Cadherin family members include desmogleins (DSGs) and desmocollins (DSCs), both of which have functions that depend on calcium, and which share homology with the classical cadherins found in AJs, also known as fascia adherens [48]. Extracellular domains of these transmembrane proteins form *trans* homo- and hetero-dimers (DSG-DSG, DSC-DSC, and DSG-DSC) [49] that join adjacent cells through direct physical interactions that span spaces between cells. Intracellularly, DSGs and DSCs bind to adaptor proteins plakoglobin (JUP or γ-catenin) and plakophilins (PKPs), both of which are members of the armadillo family. Members of this family mediate interactions between the cadherin tails of DSGs and DSCs with the plakin family member protein desmoplakin (DSP). DSP anchors the desmosome to the cytoskeleton network through specific binding to the IF DES [48]. The C-terminal region of DSP, which contains three plakin repeat domains (PRD-A, -B, and -C) mediates binding to the IFs. This property distinguishes desmosomes from AJs, which bind to actin microfilaments [50]. 

In the adult heart, the overall structure of the desmosome and its organization are similar to that found in other tissues ((1) in Figure 2A). DSG2 and DSC2 (the primary isoforms of DSGs and DSCs in the heart) bind JUP and PKP2 (the cardiac isoform of PKP), which in turn, binds DSP. DES, which binds to DSP, provides a direct connection between desmosomes and sarcomeric Z-discs. Additionally, the desmin network links Z-discs to sarcolemmal costameres and to the nuclear envelope [45]. Thus, desmosomes are physically confined to the cell membrane ((1) in Figure 2A), but their intracellular interactions extend to the cytoskeletal network which link it directly to the contractile machinery of the CM. When one component of this network is perturbed through genetic mutation ((3) in Figure 2A), a myriad of pathogenic effects potentially can occur within the cell including loss of junctional integrity ((2) in Figure 2A) and triggering or suppression of normal signaling pathways ((4) in Figure 2A).

The dependence of these junctional complexes on one another is underscored by the maturation process of the ICD, which relies on the proper initial formation of AJs, followed by assembly of the desmosomes, and the subsequent colocalization of ion channels and gap junctions to the ICD [44] (Figure 2). Developmentally, the ICD does not form mechanical junctions with an adult-like pattern until after one year of age [56]. Prior to this, the desmosome proteins are distributed mostly near the cell membrane, often in a punctate manner that is in direct contact with adjacent CMs [35]. Interestingly, this type of pattern is also evident in multicellular preparations of hiPSC-CMs (Figure 3), which exhibit a more immature, fetal-like phenotype relative to adult CMs. To better visualize a network of relevant interactions that likely occur within the cardiac ICD, we used the open-access STRING database [57] to show the interrelatedness of cellular components at the molecular level and how dACM might contribute to pathological phenotypes. For this we used a PubMed query for “intercalated disc”, which was moderately filtered for heart-tissue-specific genes with confidence scores >0.50. This analysis yielded a large network of proteins (Figure 2B) that include desmosomes, adherens junctions (*CDH2*, catenin family of proteins), and ion channels (*GJA1*, *SCN5A*), which are highlighted in a smaller, focused network in Figure 2C. Proteins involved with calcium handling (*CACNA1C, CASQ2,* and *RYR2*) and the contractile machinery of the CM (*ACTN2, TTN, MYH6/7,* and *MYL2/7*) were implicated in this network (Figure 2B). These and other non-ICD-related proteins, such as *FLNC, TMEM43, RBM20, TJP1, NKX2-5,* and *OBSCN*, overlap with some non-desmosomal genes that have been implicated in the genetics of ACM (see Figure 1). Altogether, these results illustrate the extent to which the desmosomes can influence CM morphology and physiology. 

Currently, there are three theories that help explain how desmosome disruption in heart leads to dACM phenotypes. The first relies on the innate functional role of the desmosomes in cardiac cell biology. Disruption of any component of the desmosome may lead to weakened cell-to-cell adhesion, resulting in CM detachment, which may trigger apoptosis and remodeling characteristics of myocardial injury [60,61]. Fibrofatty remodeling of the heart may subsequently serve as an arrhythmogenic substrate [62,63]. A second theory links desmosomes directly to ACM. It involves the dual signaling role of JUP and PKP2, which can perturb the homeostasis of several signaling pathways ((4) in Figure 2A). JUP, a paralogue of β-catenin, can interfere with β-catenin’s role in canonical Wnt signaling. When JUP fails to properly localize to the desmosomes, an increase in free cytosolic JUP may inhibit Wnt activity, which could lead to adipogenesis [45,51]. Alternatively, downregulation of PKP2 likely reduces Wnt signaling by promoting nuclear translocation of JUP and stimulating the Hippo-YAP pathway [44,64]. Reduction in PKP2 has also been linked to fibrosis through TGF-β1 activation [44,65], and has been associated with pro-inflammatory NFκB signaling, possibly through transcriptomic coupling of PKP2 with endogenous inflammatory and immune pathways [53]. The third possibility centers on the formation of desmosomes, and how any disruption to this process may adversely affect other components critical to ICD function (Figure 2B,C). Since the mature ICD relies on the sequential formation of components of the desmosome [44], any reduction in desmosome proteins would be accompanied by a decrease in Cx43 (GJA1) and Na_v_1.5 (SCN5A) content and/or localization [66,67,68]. Thus, proper localization and interactions among these proteins are essential for formation of a normal and functional ICD such that disturbances to these processes may result in a dACM phenotype [44]. Indeed, dysregulated expression or localization of components of the ICD, such as CDH2, ankyrin-G, or members of the catenin family of cadherin adaptor proteins, have been observed to affect the expression, localization, and trafficking of Cx43 and Na_v_1.5 [66,69,70]. Although three theories have been described, they are not mutually exclusive, and it is likely that aspects of each theory may contribute to dACM.

## 4. Desmosomal Genes and ACM

Human endomyocardial autopsy and biopsy samples, primary cells, and animal models have been pivotal in supporting the link between desmosome mutations and ACM. We describe in this section the genes and the protein structures of the desmosomes (also see Figure 2), as well as information obtained from patients, patient cells, modified cells, and animal models that have linked human ACM with mutations in genes encoding desmosomal proteins. 

### 4.1. Plakophilin (PKP)

Plakophilins, encoded by four genes (*PKP1, 2, 3,* and *4*), are located on chromosomes 1q32, 12p13, 11p15, and 2q23-q31, respectively [71]. These proteins show complex tissue-restricted expression patterns and are members of the armadillo gene family. The term armadillo was first described in Drosophila and is the historical name for the β-catenin gene, which is thought to regulate homodimerization of a-catenin to mediate actin branching and bundling. Armadillo repeat domains interact with binding partners to regulate cell-contact- and cytoskeleton-associated protein interactions and a wide range of other functions [72,73]. PKP proteins (1, 2, and 4) contain 9 armadillo repeats (repetitive amino acid sequences of approximately 42 residues) with a flexible insert between repeats 5 and 6 [71]. *PKP1* and *PKP2* each have two isoforms, one with a shorter “a” variant and another with a longer “b” variant generated by alternative splicing. PKP4 isoforms are similar to those of PKP1 and PKP2, but the two PKP3 isoforms are produced by alternative promoter usage. PKP1, 2, and 3 and their isoforms can be found in desmosomes and in the nucleus, the location of which is influenced by 14-3-3 proteins [47]. PKP2 phosphorylation at serine residue 82 by Cdc25C-associated kinase 1, for example, creates a 14-3-3 binding site that is located mainly at the plasma membrane. PKP2 mutants that cannot bind 14-3-3 are found mostly in the nucleus [74]. PKP4 is involved in Rho regulation during cytokinesis. 

The PKP2 gene, the main isoform in the human heart, encodes a protein of 881 amino acids (Figure 4A). It is especially prominent in the right ventricle [75]. Mutations in PKP2 are typically inherited and closely associated with familial forms of the disease that typically impact the right ventricle. From patient studies, Gerull et al. described nearly 25 different heterozygous mutations from individuals of Western European descent. Of these, 12 were insertion–deletion, 6 were nonsense, 4 were missense, and 3 were splice site mutations. Most of the mutations were present in the C-terminal half of the molecule [76]. In a Polish population of patients that met ITFC criteria for ACM, Biernacka et al. identified five frameshift, two nonsense, two splicing, and one missense mutations in *PKP2*. Through comparison studies, they found that patients with a *PKP2* mutation were younger at diagnosis, more often had negative T waves, had higher left ventricular ejection fracts, and were less likely to have symptoms of heart failure than ACM patients without *PKP2* mutations [77]. Individuals in the non-PKP2 group tended to have a better prognosis with less SCD or need for a heart transplant [77]. Syrris et al. investigated 100 Caucasian patients with *PKP2*-associated ACM. Nine different mutations were identified, including five that had not previously been reported in *PKP2* (R413X, A733fsX740, L586fsX658, V570fsX576, and P533fsX561); these five mutations included one which caused a premature truncation of PKP2 and four which were frameshift mutations [78]. Dalal et al. reported that males were more likely to have structural and conduction abnormalities than females [79], and the phenotypes within the tested cohort ranged from asymptomatic to severe (including SCD) [79]. Van Tintelen et al. studied ACM cases in a Dutch population, and of the 34 patients that satisfied the index criteria for ACM, 23 were familial of which 16 (70%) were caused by *PKP2* mutations [80]. *PKP2* mutations also have been associated with SCD, particularly in young athletes, and with left ventricular involvement during late stages of the disease [81]. 

The role of *PKP2* mutations in dACM has been studied in animal models and in human cells. Cerrone et al. found that loss of PKP2 in a stable, lentivirus-modified HL-1 cell line, a cell line originally derived from the AT-1 mouse atrial CM tumor lineage, resulted in reduced sodium current (*I*_Na_) relative to controls [68,94]. This study was the first to demonstrate that loss of Na_V_1.5 at the ICD could be caused by a genetic variation in *PKP2*. These *PKP2* mutations could also contribute to Brugada syndrome, which is characterized by arrhythmias associated with very fast heart rates. Interestingly, the authors speculated that the effects of *PKP2* mutations may not have been due to its function as a component of the desmosome, but rather that cytoskeletal alterations affecting ion channel trafficking may be important for this phenotype [68]. Chelko et al. showed that buccal mucosa cells cultivated in vitro from patients with *PKP2* mutations generally had a reduction in PKP1, JUP, and connexin 43 (Cx43) immunostaining; however, DSP was not affected. Interestingly, the loss of PKP1 staining in these cells was more consistent than the loss of PKP2 staining. Treatment of the cultured cells with SB216763, a glycogen synthase kinase 3-β (GSK3β) inhibitor previously reported to reverse ACM disease phenotypes in zebrafish [95], could normalize desmosome protein distributions in the cells through a process thought to involve Wnt/β-catenin signaling [96]. In zebrafish subjected to PKP2 knockdown using morpholinos, Moriarty et al. found that heart development was stifled by the presence of *PKP2* mutations [97]. It was accompanied by cardiac edema, heart looping defects/blood pooling, reduced heart rate, and changes to the desmosome structures and numbers. Importantly, this phenotype could be rescued in the zebrafish by injection of PKP2 mRNA. 

Animal models have provided additional direct evidence of the role of PKP2 in ACM. Using transgenic animal models, Grossman et al. developed a null mutation by homologous recombination in *PKP2* [76]. During development, mouse homozygote embryos developed right ventricular (RV) wall thinning, had hemopericardium at embryonic day 10 with blood aggregates in the intraperitoneal cavity that led to small holes in the lining (endothelium) of the heart and developing vasculature, and did not survive beyond ED11 [76]. Cerrone et al. reported that missense *PKP2* mutations with PKP2 deficits in CMs were correlated with sodium current (*I*_Na_) deficiency, with reduced numbers of sodium channels at the ICD and with an increased separation of microtubules from the cell end, which was defined by the midline of N-cadherin clusters. PKP2 deficiency thus affected the ability of microtubules to reach the ICD, which likely impacted the delivery to the ICD of proteins related to sodium channel function [68]. Using a stable cardiac expression mutation of *PKP2* (c.2203C>T), Cruz et al. demonstrated that the R735X mutation functioned as a dominant-negative variant that led to an exercise-dependent dACM phenotype with impaired global right ventricular systolic function and regional wall motion abnormalities [98]. In summary, these and other studies from human and mouse models show that mutations present in *PKP2* account for ~70% of all diagnoses attributable to desmosome mutations, typically affect males more than females, and are often associated with SCD in young athletes. Furthermore, PKP2 functionally plays an essential role in the development of desmosomal and ICD structures.

### 4.2. Desmoglein (DSG)

In humans, desmoglein proteins are encoded by four genes (*DSG1-4*) located as a gene cluster on chromosome 18q12. These proteins are all members of the cadherin cell adhesion molecule superfamily that mediates calcium-dependent cell–cell adhesion. This family of proteins is involved in numerous biological processes that include cell recognition, communication, and signaling as well as angiogenesis and morphogenesis. DSG genes are comprised of either 15 or 16 exons, and their expression patterns display a tissue and differentiation-specific distribution [99]. DSG1 and 3 are mostly found in stratified squamous epithelia among other tissues, while DSG4 is present mostly in human hair follicle. DSG2 is the most widely expressed isoform, and it is the only DSG protein present in CMs. It is composed of 1118 amino acids (Figure 4B), and it is a calcium-binding transmembrane glycoprotein.

It is estimated that 5–10% of cases with dACM carry a pathogenic *DSG2* mutation variant, with the majority being rare missense mutations [99]. Several nonsense, insertion, deletion, and splice site variants that lead to frameshifts and premature termination codons have been observed [100,101,102]. *DSG2* pathogenic variants often show prominent left ventricular involvement during early stages of the disease. From human studies, Pilichou et al. found nine heterozygous *DSG2* mutations in a series of unrelated dACM patients, including five missense, one nonsense, two insertion/deletion, and one splice-site mutation [101]. These *DSG2* mutations in dACM patients appeared to predominantly affect the left ventricle with desmosome structure remodeling, a loss of myocytes, and fibrofatty tissue replacement [101]. In 2021, Lao et al. published a case report of a 28-year-old woman who presented with left ventricular dominant dACM who had a heterozygous pathogenic variant in exon 15 (c.3059–3062del, p.Glu1020Alafs*18) [103]. Clinically, this patient had epicardial and midmyocardial fatty infiltration in the left ventricles, regional dyskinesis, and reduced left ventricular ejection fraction [103]. Using next generation sequencing (NGS) and single nucleotide polymorphism (SNP) arrays on samples from two individuals with ACM, Brodehl et al. reported a homozygous splice mutation (c.378+1G>T) and a nonsense mutation (p.L772X) coupled with a large deletion in *DSG2* that appeared to exhibit an autosomal recessive inheritance pattern in patients [104]. Whole exome sequencing uncovered a homozygous mutation (c.1592T>G) regulated by DNA modification that resulted in amino acid sequence changes, protein structure effects, and splice site changes in samples. The patient with this biallelic mutation had increased levels of cardiac troponin I, myocardial edema in the lateral ventricular wall and apex, and an electrocardiogram with multiple premature ventricular beats. This unusual mutation in DSG2 resulted in an initial diagnosis of pediatric myocarditis, which was subsequently classified as ACM. Due to the early onset and distinct clinical features, an expanded clinical feature spectrum of DSG2-associated dACM was developed to account for this phenotype [105]. These studies, among others, established a basis for dACM that involves DSG2 and left ventricular disease phenotypes [106,107,108].

As described above, Chelko et al. utilized buccal mucosa cells to assess dACM in patient cells. Using these human *DSG2* mutant cells, they found that PKP1 was not affected by *DSG2* mutations, in contrast to what was observed for *PKP2*; however, these mutations led to a reduction in Cx43 and JUP, which could be improved by treatment with a GSK3β inhibitor [96]. In mouse models, *DSG2*^−/−^ embryos die during early embryogenesis, while *DSG2*^+/−^ embryos often die at or shortly after implantation. In blastocysts, the distribution of DSP in the desmosomal plaque was disturbed; however, neither E-cadherin nor β-catenin appeared to be affected [109]. Pilichou et al. demonstrated the impact of *DSG2* mutations by creating a transgenic mouse line with cardiac-restricted expression [110]. Mice with a N271S mutation developed clinical features of dACM, which included aneurysms, spontaneous ventricular arrhythmias, cardiac dysfunction, biventricular dilation, and sudden cardiac death (SCD) at a young age. CM necrosis was also observed. *DSG2* mutations thus can promote myocardial damage that may lead to myocardial atrophy, inflammation, fibrofatty replacement of myocytes, and calcification [110]. Krushce et al. developed a model of dACM in mice lacking exons 4–6 (targeted deletion) of *DSG2* [111], which led to an increase in sudden cardiac death. These mice showed an increase in CM cell death and necrosis as well as fibrotic lesions. Proliferation of cells within fibrotic lesions (increased scar/fibrous tissues) were more pronounced in young versus old mice. With aging, however, these mice developed fibrous tissues and ventricular dilation, all hallmarks of dACM. Rizzo et al. studied transgenic mice that overexpressed mutant *DSG2* (N271S). This mutation was a homologue of the *DSG2* N266S mutation previously identified in an dACM patient by Pilichou et al. [101]. Rizzo et al. found that *DSG2* mutations induced widening of the ICD at the level of the area composita, a finding that coincided with slowed conduction, a reduction in action potential upstroke velocity, and a reduction in *I*_Na_. They provided further evidence of an in vivo interaction between DSG2 and Na_v_1.5 as a molecular mechanism responsible for the slowed conduction and presence of arrhythmias in dACM prior to any overt structural changes [112] Kant et al. described a transgenic mouse line with CM-restricted ablation. DSG2 protein levels in hearts were reduced to <3% of normal, and all the animals developed dACM with severe morphological alterations during the postnatal period. The phenotypes included chamber dilation, calcifying CM necrosis, inflammation, interstitial and focal replacement fibrosis, and conduction defects with an abnormal Cx43 distribution [113]. Chelko et al. reported on *DSG2* homozygous mutant mice lacking exons 4 and 5 that recapitulated dACM by 8 weeks of age compared with controls. Phenotypically, these mice had abnormal JUP and Cx43 at the ICD and extensive biventricular fibrosis but exhibited a normal distribution of CDH2 relative to controls. Inhibition of GSK3β could improve the lifespan of these mice through improvements in ventricular ectopy, function, and myocardial fibrosis/inflammation [114]. Another cardiac-restricted knockout *DSG2* model was developed [115]. Qiu et al. later showed that activation of PPARɑ provided a cardioprotective effect through its phosphorylation of STAT3 and SMAD3. This inhibited cardiac fibrosis in ACM, suggesting a possible treatment for *DSG2*-mediated dACM [116]. Altogether, these findings are consistent with *DSG2* mutations leading to ACM; however, it is noteworthy that a reduction in DSG2 by > 50% was required for overexpression of *DSG2* N271S to lead to ACM, at least in mouse models. In summary, these studies showed that DSG2 mutations often affect the left or both ventricles in humans, compromise sodium channel and connexin distributions, which promote arrhythmias, and adversely affect ICD formation.

### 4.3. Desmocollin (DSC)

Desmocollins (DSCs) are type I transmembrane glycoproteins and members of the cadherin cell adhesion molecule superfamily. These proteins are encoded by three genes (*DSC1, DSC2,* and *DSC3*) located as part of two clusters on chromosomes 18q12 [117]. Each gene encodes a pair of proteins with a larger “a” form and shorter “b” form with differences in their C-terminus. Full length DSC2 encodes a protein that is 901 amino acids in length (Figure 4C); however, gene transcripts vary in length due to two initiation sites of transcription and two splice variants. Only two protein isoforms are formed from these transcripts due to alternative splicing at exon 16. DSC3 has a very short variant of 214 amino acids that has an incomplete coding sequence. Functionally, DSC proteins provide tensile strength to the desmosomes, and they act as molecular sensors and mediators of signal transduction [118]. DSC1 and DSC3 proteins are prevalent in epidermis and some other tissues. DSC2 is the only desmocollin isoform present in CMs, and the translated protein is localized mainly to the desmosomes within the ICD (Figure 2). 

*DSC2* variants are generally rare in patients with ACM. Mutations in this gene predominantly affect left ventricular electrical activity during early stages of the disease; however, a pathogenic role of these mutations in the right ventricular form of dACM has also been consistently documented (1–2% of cases) [119]. Gehmlich et al. reported two missense (R203C and T275M) mutations, one of which led to a premature termination codon (predicted to not produce a functional protein), and another one causing a frameshift mutation (A8a97fsX900). The prematurely terminated protein likely led to haploinsufficiency, while the other led to defects in proteolytic cleavage at the N-terminal cadherin domains. As a consequence, the DSC2 proteins fail or only partially localize to the desmosomes of the ICD [120]. In a 58-year-old male who had ventricular arrhythmias with a left bundle branch block, Heuser et al. identified a heterozygous splice-acceptor-site mutation in intron 5 (c.631–2A>G) of *DSC2,* which leads to a cryptic splice-acceptor site and the creation of a downstream premature termination codon [121]. Gerull et al. reported a homozygous founder mutation in *DSC2* in two large families of the Alberta Hutterite population that led to a truncation mutation (c.1660C>T). Immunostaining of endomyocardial biopsies confirmed the presence of truncated DSC2 protein at the ICD [6]. The subgroup of affected individuals maintained a carrier frequency of 9.4% (1 in 10.6) among 1535 Schmiedeleut Hutterites [6]. Brodehl et al. reported a four-base pair (bp) *DSC2* deletion (c.1913_1916delAGAA, p. Q638LfsX647^hom^) that caused a frameshift mutation in dACM patients and resulted in a loss of heterozygosity with segmental interstitial uniparental isodisomy [122]. Through transmission electron microscopy, the ultrastructure of the myocytes from this patient displayed a widened ICD (i.e., gap) located in the left ventricular myocardium [122].

*DSC2* mutations have also been studied in human cells and animal models. The use of patient buccal mucosa cells with DSC2 mutations gave results that were remarkably similar to those observed for *DSG2*, but distinct from those reported for *PKP2* [96]. To better understand the consequence of the splice-acceptor-site mutation in intron 5 of the *DSC* gene in humans described above, Heuser et al. cloned the zebrafish ortholog of *DSC2* and used morpholinos to target the translation start site and the splice-acceptor sites for exons 6 and 11. The morpholino phenotypes showed a significant dose-dependent bradycardia with chamber dilation, pericardial edema, and abnormal cardiac contractility. Reduced desmosome plaque areas and loss of desmosomal midlines in developing embryos were also observed. Importantly, these phenotypes could be rescued through co-injection of the morpholinos with wild-type human DSC2 mRNA, but not by injection of mutant human DSC2 mRNA [121]. In transgenic mice with cardiac-restricted overexpression, tagged DSC2 proteins localized primarily to the ICD; however, some remained cytoplasmic. These mice developed severe myocardial necrosis, as well as fibrosis and calcification of both ventricles and the septum. An acute inflammatory response involving chemokine, cytokine, or toll-like receptor signaling, as well as macrophage infiltration into the myocardium were also reported [123]. Neither *DSC2* heterozygous (+/G790del) nor homozygous (G790del/G790del) mice showed structural or functional defects in the right ventricle or developed lethal arrhythmias. Only at six-months of age did the homozygous mutant mice show modest left ventricular dysfunction with decreased cell shortening and prolonged intracellular Ca^2+^ transients. Spontaneous Ca^2+^ transients were also observed in response to isoproterenol [124]. These results in human and animal models provide direct evidence that DSC2 mutations can lead to disrupted ICD and DSC2 proteins that are linked with dACM phenotypes. More recently, Pohl et al. investigated the in vitro impact of prodomain variants of the DSC2 protein. Prodomains and conserved positions within the prodomain are thought to be important for the subcellular transport of DSC2 to the plasma membrane. In this study, hiPSC-CMs were transfected with prodomain variants and immunostained with wheat germ agglutinin, α-actinin, DSC2, calnexin (endoplasmic reticulum), and N-acetylgalactosaminyltransferase 2 (Golgi apparatus). Variants (p.D30N, p.V52A/I, P.G77V/D/S, p.V79G, and p.I96V/T) were analyzed via confocal microscopy for their plasma membrane localization. The imaging revealed that variants P.G77V/D/S and p.V79G were expressed but did not localize to the plasma membrane. Instead, these two variants remained in the endoplasmic reticulum or Golgi apparatus, showing that prodomain variants in DSC2 could prevent subcellular transport to the plasma membrane [125]. In summary, *DSC2* mutations in humans affect left and right ventricles, and from animal model studies, desmosomal plaques are reduced, mice develop necrosis, fibrosis, and calcification of both ventricles and the septum, and calcium handling is affected. 

### 4.4. Desmoplakin (DSP)

Desmoplakin (*DSP*) is encoded by a single gene located on chromosome 6p24 that gives rise to two isoforms: DPI and DPII, which are composed of 2871 (Figure 4D) and 2272 amino acids, respectively. DPI is the predominant form present in the heart (Figure 2). The protein functions as a homodimer with a dumbbell-shaped conformation [126]. Unique to this desmosomal protein is the presence of a “plakin domain”, comprising six spectrin repeat domains separated by SH3 domains. The plakin family consists of proteins characterized by a multimodular structure that enables them to function as cross-linkers of the cytoskeleton (microfilaments, microtubules, and intermediate filaments). This fosters cytoskeletal component interactions with each other and with junctional complexes (adhesion molecules) on the plasma membrane to help control cell shape and polarity through modulation of cytoskeletal dynamics. These domains also help regulate processes like cell adhesion, migration, polarization, or signaling pathways. The N-terminal globular head domain is composed of a-helical bundles required for localization of the protein to the desmosome, where it interacts with the N-terminal region of PKP2 and with the C-termini of DSC and DSG2 (Figure 2). The middle region of DSP contains a coiled-coil rod domain responsible for homodimerization. The C-terminal is composed of three plakin repeat domains (A, B, and C) required for cytoskeletal component binding. 

*DSP* pathogenic variants have been associated mostly with left ventricular dominant dACM; however, recessive forms have also been described. The first reported cases of autosomal recessive *DSP* mutation were by Kaplan et al. They identified a homozygous point mutation (7901delG), which led to a premature stop codon and truncation of the protein in the C-terminal portion of the molecule [127]. This truncation caused generalized striated keratoderma and dilated left ventricular CMP that ultimately evolved into biventricular cardiomyopathy. Histological assessments of a patient’s heart revealed a unique CMP characterized by ventricular hypertrophy with dilatation, ultrastructural abnormalities of ICD with decreased amounts of DSP, JUP, and Cx43, and a failure of desmin to localize to the ICD. There was, however, no evidence of fibrofatty infiltration. Autosomal dominant forms of DSP mutations in dACM have also been described. Rampazzo et al. identified a mutation (S299R) in exon 7 of DSP that modifies a putative phosphorylation site in the N-terminal domain binding JUP [128]. This mutation resulted in biventricular dilation associated with woolly hairs and palmoplantar keratoderma [128]. Bauce et al. subsequently reported two novel missense mutations (R1775I, R1255K) and one intron–exon splicing region mutation (c.423-1G>A -intron 3) in four families that demonstrated left ventricular involvement and a high occurrence of sudden cardiac death (SCD) [129]. Patients experienced myocardial enzyme release and chest pain with ST segment elevation in their electrocardiograms and had a mutation in *DSP* that was identified as c.423-1G>A. Sen-Chowdhry et al. later published a cohort study of 200 patients with dACM probands. These individuals underwent cardiac magnetic resonance imaging, which indicated high prevalence of left ventricular involvement with dACM and a higher rate of ventricular arrhythmias [130]. 

*DSP* mutations in modified cell lines and transgenic mouse models have established the role of this protein in ACM. Garcia-Gras et al. found that *DSP* null mice experienced early embryonic lethality [131] as did transgenic lines with cardiac deletion of *DSP* [51]. Although most cardiac null *DSP* mice showed growth arrest at E10-E12, some *DSP^−/−^* mice survived the embryonic period but died, usually within the first 2 weeks post-partum. Heterozygote adults experienced premature death, displayed enlarged cardiac chambers, and had poorly organized myocytes with large areas of patchy fibrosis. Fat droplet accumulation was also observed predominantly at the site of fibrosis. Target genes of canonical Wnt/β-catenin signaling were decreased in these animals, while transcripts for adipogenic genes (C/EBPa and adiponectin were increased). Garcia-Gras et al. also utilized HL-1 cells expressing siRNAs against DSP. Suppression of DSP transcripts led to nuclear localization of JUP and a two-fold reduction in canonical Wnt/β-catenin signaling. The cells had increased transcripts for adipogenic and fibrogenic genes, as well as accumulation of fat droplets. Huang et al. utilized Myh6-Cre mice to selectively delete *DSP* in CMs. These mice showed growth arrest at ~E11.5, similar to prior observations by Garcia-Gras et al. [51,132]. Stevens et al. found that homozygote *DSP* (R452G) mice died during embryonic development prior to E10, whereas heterozygote mice with this mutation survived. Adult mice did not display any changes in cardiac structure or function, nor did they have baseline arrhythmias or electrical abnormalities in the absence of stress. However, with pressure overload induced by transverse aortic constriction, the mutant mice progressed to heart failure more quickly than normal. These mutant mice exhibited chamber dilation with reduced fractional shortening. They also developed more severe T-wave inversions and displayed a fragmented pattern in the QRS complex. Catecholaminergic (epinephrine) challenge resulted in an increased prevalence and severity of arrhythmias, which included ventricular tachycardia, bigeminy, a higher risk of atrioventricular block, and premature ventricular contractions. Morphologically, these animals showed altered Cx43 localization, which were disrupted further following stress. These findings highlight the role of cardiac stress in the development of dACM disease phenotypes [133]. Finally, another study demonstrated the role of *DSP* in Carvajal-Huerta syndrome. A frameshift mutation at 38,288,978 bp of chromosome 13 in the *DSP* gene led to a C-terminus truncation and a phenotype consistent with this syndrome and caused abnormal ruffled hair, epidermal blistering, abnormal ECGs, and ventricular fibrosis [134]. In summary, patients with *DSP* mutations have clinical features that include arrhythmias of left ventricular origin, T wave inversions, and prominent regional myocardial fibrosis in the left ventricle as well as autosomal-dominant and recessive inheritance traits. The animal models also show defects in desmosome protein localization, intracellular signaling cascades, electrical abnormalities, and enhanced fat deposition. 

### 4.5. Plakoglobin (JUP)

Plakoglobin (*JUP)* was one of the first desmosomal genes reported to be associated with dACM in human patients. It is encoded on human chromosome 17q21 and translates into a cytoplasmic protein of 745 amino acids. JUP is a member of the armadillo superfamily, and it contains 12 armadillo repeats, flanked by N- and C-terminal domains. Functionally, it links cadherins to the actin cytoskeleton, is essential for the normal development of ICD (Figure 2), and influences the arrangement and function of the cytoskeleton and the arrangement of cells within a tissue. It is the only protein known to be a common constituent of sub-membranous plaques. JUP forms links in these plaques with IFs and may contribute to desmoplakin and desmosomal cadherin protein interactions. JUP also may be important for cross-talk between AJs and desmosomes [135,136].

The number of patients that suffer from *JUP* mutations is relatively small. A homozygous deletion was identified via genetic linkage analysis to be the cause of the autosomal recessive Naxos disease. Specifically, a two-base pair (bp) deletion in *JUP* on chromosome 17q21 (PK2157del2) led to a premature stop codon [137] with loss of myocyte integrity and junction disruption, leading to cell death and fibrofatty replacement. Like Carvajal-Huerta syndrome, these mutations led to cardiocutaneous phenotypes characterized by woolly hair and palmoplantar keratoderma. Histologically, the hearts had a typical pattern of dACM with fibrofatty replacement of the right (mostly subepicardial and midmyocardial layers) and left ventricles. Surviving myocytes were surrounded by fibrotic tissue and embedded within fatty tissue and Cx43 was reduced at the ICD. Autosomal dominant mutations for *JUP* have also been described. Asimaki et al. reported a *JUP* mutation at S39_K40insS in a patient of German descent. From a right ventricular biopsy, they observed reductions in plakoglobin localization, JUP, and Cx43 [138]. Using human embryonic kidney (HEK) cells and transfection of an expression vector containing the mutation, they reported that, relative to controls, the transfected HEK cells showed increased proliferation, displayed lower rates of apoptosis, and had reduced numbers and sizes of desmosomes. These data suggest that this mutation disrupted the mechanical integrity of cells. Lui et al. reported a 24-year-old male admitted to the hospital for syncope during sports activity who had an electrocardiogram with inverted T-waves [139]. Sanger sequencing revealed an autosomal dominant mutation of c.1729C>T/p.R775C of *JUP*. Using AC16 human cardiomyocytes, derived from the fusion of primary cells from human ventricle with SV40 transformed uridine auxotroph human fibroblasts, a mutant AC16 CM cell line (R577C) was developed and compared with control cells. Their results indicated decreased expressions of DSG2 and Cx43, which was speculated to contribute to the disruption of desmosomes and intermediate junction stability. 

Plakoglobin animal models have been instrumental to our understanding of how mutations in this gene promote dACM. Ruiz et al. showed that *JUP* null mice die between E10 and E16 due to cardiac defects [135]. Pericardial cavities of mutant embryos were often swollen, filled with blood, and the heart walls were often ruptured, but still contracting. Histologically, the ICD was grossly altered, with typical desmosomes no longer detectable. In place of the desmosomes, AJ plaques were prominent. Interestingly, typical desmosomes were present in epithelial organs, suggesting that JUP is essential for the normal development of myocardial desmosomes and proper formation of the ICD. In a study on ten-month-old heterozygous *JUP*-deficient mice compared with wild-type siblings, Kirchoff et al. showed that mutant mice exhibited increased right ventricular volume, reduced right ventricular function, and spontaneous ventricular ectopy [140]. There was no reported change to size and function of the left ventricle in the affected mutants. Isolated, perfused mutant hearts demonstrated spontaneous ventricular tachycardia and had prolonged right ventricular conduction times. Fabritz et al. studied the effects of endurance training (7 weeks of daily swimming) on wild-type and littermate heterozygous plakoglobin-deficient mice [141]. Mutant mice demonstrated right ventricle enlargement relative to the wild-type mice. Mouse hearts were subsequently isolated and perfused, and these hearts exhibited ventricular tachycardias and reduced right ventricular longitudinal conduction velocity. Histology on the excised hearts revealed reduced myocardial JUP but no changes to CDH2. In a zebrafish *JUP* knockdown model, Martin et al. developed and implemented a morpholino against the AUG region common to plakoglobin-1a and 1b. Zebrafish with the mutation demonstrated decreased heart size, reduced heartbeat, cardiac edema, and reflux of blood between the developing heart chambers. Lombardi et al. reported the overexpression of a truncated form of *JUP* (23654del2) [142]. In this mouse model, truncated *JUP* mutants demonstrated fibrofatty development, cardiac dysfunction, and premature death. Cardiac progenitor cells isolated from mutant mice were found to have increased adipogenesis, increased adipogenic factors, and reduced levels of adipogenesis inhibitors. Li et al. reported the development of mice with CM-restricted ablation of *JUP* [143]. Plakoglobin-deficient mutants experienced SCD as early as 1 month of age and typically had an average lifespan of 4.6 months. Mutant hearts also had many of the features observed in the clinic, including ventricular dilation, cardiac fibrosis, cardiac dysfunction, ventricular aneurysm, and spontaneous ventricular arrhythmias. These data are all consistent with *JUP*, like the other desmosomal proteins, having a critical role in the formation of desmosomes and ICD in myocardial tissues, an influence on cardiac electrical conduction, and a disease phenotype restricted, at least initially, to the right ventricle.

### 4.6. Desmin

Desminopathies, caused by mutations in DES, represent one of the most common IF human disorders, some of which have been linked directly to ACM [12,144,145,146,147]. It is located on chromosome 2q35 and is prevalent in heart and in skeletal muscle. It is not, however, typically considered to be part of the desmosome (Figure 1) [146,148], but we include a short discussion of this type III IF protein in ACM since it binds DSP and links the desmosomes to numerous intracellular CM components, such as the myofibril Z-discs, cell nucleus, mitochondria, and several other organelles [149,150,151,152]. One of the earliest discovered *DES* mutations linked to ACM is a missense N116S mutation, which resulted in disturbance of the aggresomes of skeletal muscle fibers, possibly due to impaired desmin filament formation [144]. In a meta-analysis of 159 patients with various *DES* variants, 60% of the cohort experienced cardiac conduction disease or arrhythmias with atrioventricular block [153]. Bermúdez-Jiménez et al. found that the largest known family with a p.E401D mutation in this gene primarily exhibited familial ACM, suggesting that the prevalence of mutant *DES* ACM may be underestimated. Histological and molecular analyses of these tissues revealed abnormal cell growth, reduced desmin mRNA, and disrupted ICD [146], the last of which is suggestive of dACM. Further insights into the pathogenesis of desminopathies and, by extension, desmin-related ACM, have been made possible through the development of a knockout *DES* murine model, which many laboratories have used to recapitulate desmin-related CMP phenotypes [154]. Using a desmin null mouse model, researchers have identified a link between a compromised desmin network and dysregulated mitochondrial function that leads to oxidative stress and cell death [155].

## 5. Human-Induced Pluripotent Stem Cells to Model dACM 

A current challenge limiting advances in the study of dACM is the limited availability of human tissues and suitable human cells for investigation. Although animal models have been informative, there are profound functional differences in cardiac electrophysiology between mice and humans [156]. As an alternative, hiPSCs differentiated into CMs represent a powerful tool for modeling dACM in vitro and have numerous advantages over other mammalian systems for the cellular study of development, physiology, and disease [157]. The in vitro differentiated cells generated from hiPSCs have the same DNA as the patient from whom they were derived, and advances in hiPSC technologies have led to improved understanding of the genetic factors that contribute to arrhythmic syndromes [157,158,159,160,161], valvular and vascular disorders [162,163], and metabolic risk factors that contribute to CMPs and ischemic heart disease [164,165,166,167]. With gene-editing techniques like CRISPR-Cas9, iPSCs with gene variants can be corrected to obtain isogenic controls, which are useful to remove confounding effects arising from the presence of unknown silent pathological mutations in unrelated control cell lines [168,169,170,171]. Similarly, a specific mutation of interest can be artificially introduced via CRISPR-Cas9 into iPSCs from healthy patients [172]. This approach can be particularly useful when validating mechanistic links between genotype and phenotype or when evaluating contributing factors that may affect penetrance, such as genetic background or variable expression levels [173]. Human iPSC-derived CMs (hiPSC-CMs) thus represent a reproducible and scalable model system for basic cell biology. 

As dACM is fundamentally a disease of the myocardium, we illustrate the utility of these cells for studying desmosome localization through immunostaining of normal and mutant hiPSC-CMs (Figure 3). Relative to control cells, we find that hiPSC-CMs harboring either a PKP2 or a DSG2 mutation likely have compromised desmosomes and, by extension, ICD integrity. This is evident in the differential pattern of immunosignal intensity and/or distribution between cells, not only for the protein directly affected by the mutation, but also ancillary proteins that depend on proper anchorage to the desmosome (Figure 3). As shown in Figure 3, DSC2 (first column) immunosignals are preserved in both PKP2^mut^ and the DSG2^mut^ hiPSC-CMs relative to control hiPSC-CMs. DSG2 (second column) localization to the membrane, however, is reduced in both mutant lines. This has led us to speculate that DSG2^mut^ causes a trafficking loss of DSG2 while PKP2^mut^ may have decreased anchoring of DSG2 at the membrane due to loss of membrane-localized PKP2. In both mutant hiPSC-CMs, JUP (third column) shows increased cytoplasmic immunostaining, suggesting loss of binding to mutant proteins and dislodgment from the membrane. PKP2 (fourth column) exhibits similar disruption in DSG2^mut^ hiPSC-CMs and even more severe disruption in PKP2^mut^ hiPSC-CMs, where there appears to be a complete absence of PKP2 signal along some lengths of the membrane. DSP (fifth column) signals appear reduced or incoherently expressed at cell-to-cell interfaces in both mutant hiPSC-CMs. For both PKP2^mut^ and DSG2^mut^ hiPSC-CMs, the larger areas of non-punctate N-cadherin (sixth column) signals suggest a widening of gaps between interfacing cells possibly due to dysfunctional desmosomes. Cx43 (seventh column) is present in all cell lines, although it does exhibit a heterogeneous distribution pattern between adjacent mutant hiPSC-CMs. Indeed, these images show that hiPSC-CMs are able to fundamentally recapitulate abnormal desmosome formation consistent with previous observations in human samples, mouse models, and cell culture systems. 

To further illustrate the value of hiPSC models of ACM, we will summarize the literature in this section to describe how hiPSC models have recapitulated specific phenotypic traits of dACM and how the results have improved our insights into the disease process. Although the number of patient-derived iPSC model systems of dACM is currently limited to seven disease-associated genes (PKP2, DSG2, DSC2, DSP, SCN5A, TMEM43, and OBSCN) of which the first four encode desmosome proteins, the information provided illustrates the value of this human-based system to study dACM and to reveal new mechanistic insights into its development. A schematic overview of those mutations in desmosome genes studied in hiPSCs and their effects on the protein structures is presented in Figure 4A–D. Missense variants (blue), which result from single base substitutions, do not alter the length of the protein product, although mutant proteins may undergo misfolding which may affect tertiary structure. Truncating variants (red), which typically result from base insertions, deletions, or alternative splicing, lead to partially translated wild-type protein, beginning from the N-terminus and prematurely ending at the exact site of the amino acid switch (e.g., *DSG2* p.R119X) or continuing up to 60 amino acids downstream from the mutation site (e.g., *PKP2* p.D50SfsX110). Currently, most hiPSC-CM studies focus on *PKP2* mutants, with a greater proportion of these reports investigating truncating variants over missense variants (Figure 4E).

### 5.1. PKP2 Mutant hiPSC Lines

The first successfully reprogrammed hiPSCs generated from an ACM-symptomatic patient’s fibroblasts and differentiated down the cardiac lineage were reported by Ma et al., Caspi et al., and Kim et al. in 2013 [8,58,83]. Ma et al. generated hiPSCs to test the hypothesis that CMs containing a *PKP2* mutation (c.1841T>C) would exhibit altered localization of desmosome proteins at the ICD and have a greater potential for adipocytic changes relative to control CMs. They found that PKP2 and JUP RNA and protein were reduced in hiPSC-CMs relative to controls but did not find any differences in the abundance of DSP, Cx43, and CDH2 in the area composita. Lipid droplets were also elevated in dACM iPSC-CMs relative to normal controls, and the total amount of these droplets could be increased by cell exposure to adipogenic medium. Using a different line of hiPSCs with one *PKP2* mutation (c.972InsT), Caspi et al. showed that PKP2 RNA and protein were reduced in mutant hiPSC-CMs relative to controls. JUP and Cx43 immunostainings were lower in the PKP2 mutant lines and desmosome structures were distorted (widened desmosomal gaps and extracellular spacing between cells). Our results are similar to these findings (Figure 3, denoted as PKP2^mut^). We also note additional disruption of desmoglein-2, desmoplakin, and N-cadherin in these same cells (Figure 3, see caption for details). Electrophysiologically, the *PKP2* mutant line (c.972InsT) described by Caspi et al. had a prolonged field potential rise time. Caspi et al. also reported an increase in lipid droplet formation in their mutant hiPSC-CMs that was associated with an upregulation of the transcription factor peroxisome proliferator-activated receptor-Ɣ (PPARƔ), as well as increased CM apoptosis. Kim et al. showed that heterozygous PKP2 (c.2484C>T) mutant hiPSC lines could differentiate into CMs that had abnormal JUP nuclear translocation with decreased β-catenin activity under cardiogenic conditions. Using a five-factor cocktail that enhanced PPARɑ-dependent metabolism and activated PPARƔ, they demonstrated that mutant PKP2 iPSC-CMs had exaggerated lipogenesis and increased apoptosis relative to normal CMs. Akdis et al. subsequently showed, using these same mutant cells (c.2484C>T), that lipogenesis and apoptosis worsened in response to testosterone but improved under exposure to estradiol. These results are consistent with the higher incidence of AC seen in males [174]. Kim et al. also demonstrated that homozygous PKP2 mutant lines had calcium-handling deficits. After induction with their adipogenic five-factor medium, intracellular calcium transient decay was prolonged in *PKP2* mutant hiPSC-CMs relative to controls. They also reported a significant reduction in SERCA and NCX1 mRNA levels in these cells [8]. Wen et al. subsequently utilized two published hiPSC lines from dACM patients (*PKP2*: c.2484C>T [175] and c.2013delC [176]), coupled with the three- or five-factor protocols of Kim et al. to drive metabolic maturation through induction of PPARɑ and PPARƔ. They showed that dACM hiPSC-CMs had exaggerated CM lipogenesis, apoptosis, reactive oxygen species (ROS) production, and elevated fatty acid oxidation flux relative to normal cells. Use of a PPARƔ antagonist (GW9662 or T0070907) rescued the dACM pathologies, while addition of ROS scavengers (NAC and ascorbic acid) reduced CM apoptosis [177]. In summary, these groups demonstrated that an in vitro hiPSC-CM model of dACM harboring different *PKP2* mutations could recapitulate key features of human disease that include a decrease in desmosomes, a distorted desmosome structure, increased lipid droplet formation, altered cell coupling, and enhanced apoptosis. The most pronounced effects appeared to be in hiPSC-CMs that were induced to have a more adult-like metabolism. These data provided the first evidence that hiPSC-CMs could recapitulate key traits of dACM in vitro. 

Several subsequent studies involving different *PKP2* mutations have corroborated and expanded on these early findings. Dorn et al. postulated that disturbances to the mechanical network of ICD, such as those arising from ACM-related desmosome dysfunction, could drive transcriptional changes that could account for the tendency of ACM-associated CMs to develop an adipogenic phenotype [87]. Using dACM hiPSCs carrying a heterozygous frameshift PKP2 mutation (c.1760delT), they studied cell–cell contacts at the ICD to identify how mechanosensory-regulated pathways may affect dACM phenotypes. They observed normal levels of PKP2 mRNA in the mutant cells; however, PKP2 protein levels were reduced by ~50% relative to normal cells. PKP2 levels were also decreased at the plasma membrane, and the desmosome structure appeared thin with desmosomal proteins distributed in a punctate pattern, consistent with an early developmental phenotype [35]. With time, increased deposition of lipid droplets and disrupted sarcomere structures (cTnT staining) were observed. They also found that cell–cell contacts at the ICD are essential for actin cytoskeleton remodeling, a process normally regulated by the RhoA-ROCK signaling pathway. Activation of this pathway proved essential to maintain myocardin-related transcription factor/serum responsive factor (MRTF/SRF) transcriptional activation and maintenance of CM identity. They suggested that dysregulation of this mechanosensory pathway and specifically MRTF/SRF-mediated gene activation could activate an ectopic fat gene program that could account for the formation of fibrofatty structures in myocardium.

Three additional studies with *PKP2* mutants were performed using bioengineered substrates to create more physiologically relevant model systems [85,178,179]. In the first, Martewicz et al. (2019) utilized a homozygote mutant *PKP2* hiPSC line (described above) to culture in vitro differentiated CMs on stretchable patterned substrates designed to mimic a functional cardiac syncytium. RNA abundance and the intracellular distribution of PKP2, JUP, and Na_V_1.5 differed between mutant and normal hiPSC-CMs. Subsequent application of cyclic uniaxial stretch to the cells for 60 minutes led to a conserved, canonical mechanical-stress response in normal cells. However, the mutant ACM-CM had differentially expressed genes characterized by either matrix proteins (collagens and fibronectin) or extracellular matrix-interacting proteins. Significant dysregulation in gene ontology categories for cell–cell communication and adhesion were also observed. ACM-CMs displayed a profibrotic gene expression program that was thought to account for some of the pathological phenotypes previously reported by other investigators. They concluded that ACM-CMs had an impaired mechanosensitivity that resulted from dysfunctional desmosome components. In the second, Blazeski et al. (2019) utilized engineered heart slices (EHSs) to develop a confluent, multilayered syncytium that exhibited coordinated and spontaneous contractions that could be electrically paced. ACM-CMs cultured under these conditions had dense and highly aligned sarcomeric structures. However, SCN5A transcripts were decreased relative to normal cells concomitant with an increase in PPARƔ. The mutant EHS had shorter action potentials and re-entrant arrhythmias which could be induced by S1-S2 pacing, suggesting that the microenvironment enhanced the disease phenotype of these cells. The EHS also enhanced the syncytial alignment of hiPSC-CMs for improved functional studies [180]. In the third report using bioengineered platforms, Zhang et al. (2021) coupled hiPSC-CMs with CRISPR gene editing and developed a model of dACM with *PKP2* truncating variants (PKP2tvs) and isogenic controls. Their engineered platform consisted of micropatterned substrates with rectangular ECM-coated islands that allowed individual cells to elongate. The PKP2tv-mutant lines cultivated in this manner had aberrant RNA levels and disturbed localization of junctional proteins (desmosome and AJ proteins), as well as altered electrical characteristics. The authors reported that PKP2tv-CMs had impaired cardiac tissue contractility, analogous to the systolic dysfunction observed in dACM patients. Unexpectedly, individual dACM hiPSC-CMs generated higher systolic stress and work than normal hiPSC-CMs, a result that was interpreted to mean that the mutant lines did not cause systolic dysfunction through a loss of contractility at the cellular level. Instead, they proposed that PKP2tvs-CMs impair contractility under conditions that require cell–cell adhesions. They also reported that a smaller pool of N-cadherin is available for AJ remodeling in monolayers of PKP2tv-CMs compared with normal hiPSC-CMs. This reduced pool of N-cadherin (mostly in AJs) may permit substantially higher cell–cell boundary movement or shear between CMs than would be possible in normal monolayers. They also showed that *PKP2* truncations impaired force generation in a multicellular monolayer syncytium secondary to cell–cell junction destabilization, which led to the disruption of sarcomere content, stability, and structure.

Based on prior studies implicating nuclear factor-kappa B (NF-kB) signaling in ACM-related myocardial inflammation, Chelko et al. found that mutant *PKP2* hiPSC-CMs (c.2013delC) relative to control CMs accumulated phospho-RelA/p65 in the nucleus of these cells, consistent with NF-kB signaling activation [181]. Activation occurred in response to an overall increase in inflammatory cytokines that included the pro-inflammatory mediator interleukin (IL)-1β and neutrophil chemoattractant LIX among others, being produced and secreted by the mutant hiPSC-CMs. Interestingly, these results mirrored the changes in a cytokine profile assayed in dACM mouse hearts. All together, these findings confirmed that an inflammatory response is at least, in part, initiated by CMs themselves, and that its activation is a characteristic of dACM [181]. Inhibition of NF-KB signaling prevented the inflammatory response in the mutant *PKP2* hiPSC-CMs [181], showing that this pathway represents a potential therapeutic target for the treatment of patients with dACM. 

A role for the GSK3β-Wnt/Β-catenin signaling pathways in the molecular pathogenesis of ACM was established by Khudiakov et al. They assessed the electrophysiological and signaling properties of two *PKP2* mutant hiPSC lines (c.354delT and p.Lys859Arg) and found a significant reduction in *I*_Na_ without changes in the gating properties of Na_v_1.5. Action potential upstroke velocity was also significantly decreased in the mutant CMs. The findings of reduced sodium current were consistent with previous studies performed using modified HL-1 cells [68]. This reduction in current could be restored by exogenously applied, normal PKP2 proteins. The reduction in *I*_Na_ did not seem to be from a reduced expression of *SCN5A*. Instead, the authors speculated that abnormalities of the sodium channel microdistribution in the cell membrane were associated with impaired desmosome structures. Treatment of the hiPSC-CMs with glycogen synthase kinase 3 β (GSK3β) inhibitors activated Wnt/β-catenin signaling and restored *I*_Na_ density. Moreover, a significant decrease in Wnt/β-catenin pathway activity in the mutant lines was reported relative to normal CMs, concomitant with an increase in JUP content in the cytoplasm and nucleus. 

Knowing that regional ventricular dysfunction is one of the modified Task Force diagnostic criteria for dACM [37], Inoue et al. studied a *PKP2* mutation (c.1228 dupG, p.D410fsX425) versus isogenic controls to study CM contractility and desmosome assembly [86]. They found that PKP2 deficiency leads to a 50% reduction in PKP2 desmosome assembly and to a reduction in desmosome cadherins at the cell surface of hiPSC-CMs. They further showed that *PKP2* mutant hiPSC-CMs heterozygous for the variant did not affect the abundance or localization of any other desmosome or examined ICD proteins; however, an artificially introduced homozygous frame shift showed marked reductions in membrane PKP2, DSG2, DSC2, JUP, DSP, and CDH2. Contractile dysfunction was observed in these mutant hiPSC-CMs as was a significant decrease in *CDH2* expression [86]. In contrast to the findings of Zhang et al. (2021), no changes in the localization or expression of actin or a-actinin were reported in cells harboring either one or two truncated copies of the PKP2 protein. Instead, desmosome cadherins in the outer dense plaque appeared to be prone to instability, which could lead to disrupted cell–cell adhesions. This phenotype was likely exacerbated under enhanced contractile tension and accompanied by progressive conduction disturbances. Replacement of PKP2, using adeno-associated virus-mediated gene expression, suppressed the contractile dysfunction observed in the mutant lines [86]. 

### 5.2. DSG2 Mutant hiPSC Lines

*DSG2* mutants are much more rarely associated with ACM (Figure 4B); however, mutations in this gene are usually associated with defects in the left ventricle. In 2018, El-Battrawy et al. characterized the effect of a *DSG2* mutation (p.Gly638Arg) on several cardiac ion channels [89]. They reported reduced abundance of transcripts encoded by *SCN5A* (voltage-sensitive sodium channel Na_v_1.5), *KCNN3* (small conductance calcium-activated potassium channel SK3, also known as K_Ca_2.3), and *KCNJ11* (ATP-sensitive potassium channel Kir6.2) in mutant DSG2 hiPSC-CMs. Changes to the cardiac action potential (AP) were also observed, with mutant cells exhibiting lower AP amplitude and slower AP upstroke velocity. However, resting membrane potential and AP duration (APD) were preserved [89]. Patch clamp analyses revealed reduced *I*_Na_ in the mutant cells, consistent with the slowed AP upstroke. Examination of other ion currents revealed altered *I*_NCX_, *I_t_*_o_, *I*_SK_, *I*_KATP_, and *I*_kr_, all of which can affect APD [89]. Paradoxically, for these cells the net sum of the changed currents resulted in an APD that was not significantly different from the control [89]. Nonetheless, subtle dysregulation in individual ion channels even in the absence of a change in APD can still increase arrhythmia vulnerability, given their varying effects on excitability, refractoriness, and conduction in the myocardium. Consistently, when challenged by β-adrenergic receptor agonists (e.g., epinephrine or isoproterenol), mutant DSG2 hiPSC-CMs exhibited greater APD shortening than did control CMs. The mutant cells also had more arrhythmogenic early afterdepolarization (EAD)- and delayed afterdepolarization (DAD)-like events as well as epinephrine-induced arrhythmias compared with the control cells [89].

A follow-up study by the same lab on the same mutant *DSG2* lines examined the role of nucleoside diphosphate kinase B (NDPK-B) and SK4 channels on the arrhythmogenic phenotype of dACM [182]. Prior work has shown that SK4 channels influence the pacemaker activity of cells [183], while NDPK-B exerts an enhancing effect on these channels through phosphorylation [184]. With this context, Buljubasic et al. reported upregulation of both NDPK-B and SK4 in mutant *DSG2* hiPSC-CMs relative to control cells [182]. These changes were associated with increases in cell automaticity and arrhythmic events detected by whole-cell patch clamp, findings which were recapitulated when recombinant NDPK-B was introduced into control hiPSC-CMs. Interestingly, the aberrant electrical effects could be negated through the application of PHP-1, an agent which counteracts the phosphorylation of SK4 by NDPK-B. Ultimately, this study uncovered SK4 channels as a potential therapeutic target for treating ACM-related arrhythmias.

Shiba and colleagues studied a *DSG2* variant (c.C355T, p.R119X) which was homozygous in the donor and corrected in one allele to a normal variant through homology-directed repair (HDR) [90]. Compared with HDR-generated heterozygous hiPSC-CMs, patient-derived homozygous hiPSC-CMs demonstrated reduced protein abundance and localization of DSG2 and DSC2 with no changes apparent in the other desmosomal proteins. Homozygous mutant *DSG2* cells exhibited aberrant excitation, reduced conduction velocity, arrhythmic events, tissue fragility, weak microforces, disrupted desmosomes, and disorganized myocardial fibers. Intriguingly, each of these phenotypes were significantly improved in heterozygous hiPSC-CMs. Moreover, microforces could be remediated through adeno-associated virus-mediated replacement of DSG2 in the homozygous mutant cells. These findings suggest that, for this mutation, haploinsufficiency of *DSG2* may not significantly affect cardiac function. This supposition was supported by the absence of cardiac abnormalities in both patient’s parents who harbored heterozygous mutations of this gene.

Hawthorne et al. investigated a novel *DSG2* mutation (c.2358delA, p.Asp787fs) in multicellular monolayers of hiPSC-CMs [59]. Molecular assays showed reduced DSG2 mRNA and protein localization to the cell membrane in mutant cells, similar with our immunostaining of these cells (Figure 3, denoted as DSG2^mut^) and no significant changes to other desmosomal proteins, although mild to moderate disruption of plakoglobin, plakophilin-2, and desmoplakin as well as of ICD proteins, N-cadherin (CDH2) and Cx43, can occur (Figure 3). These cells also contained thinner myofibrils, reduced sarcomere Z-line lengths, and disorganized contractile machinery, phenotypes which are consistent with some of the major CMPs described previously. Relative to controls, CMs from this mutant DSG2 line had a shortened APD, which was attributed to an upregulation of KCNQ1 transcripts. Interestingly, no changes in AP upstroke or conduction velocity were observed, although AP upstroke heterogeneity was greater in mutant CM monolayers. The authors noted that contrary to several previous reports in mutant DSG2 and mutant PKP2 hiPSC-CMs [84,89,90], the transcript abundance of SCN5A was upregulated in these cells. They also reported differential expression of major calcium-handling genes *RYR2*, *CAMK2A*, *CASQ2*, and *SLN*, which were consistent with the reduced time-to-peak calcium and altered calcium decay kinetics observed in the intracellular calcium transients. On the other hand, complete DSG2 suppression through knock-down experiments in control cells resulted in reduced AP upstroke velocity, conduction slowing, and no changes to AP upstroke heterogeneity or calcium handling, unlike what was observed in the mutant DSG2 cells, although a similar result of shortened APD was obtained [59]. These findings are in line with the findings by Shiba et al., whose data suggest that a complete lack of wild-type DSG2 may have different or more severe effects on CM electrophysiology compared with DSG2 haploinsufficiency [90]. Finally, following the work of Chelko et al. [182], the group also found abnormal inflammatory signaling in their mutant DSG2 CMs. Specifically, cytokines implicated in the innate immune response, in chemoattractant and pleiotropic regulation, and in regulation of immune cell differentiation and proliferation were found to be upregulated in mutant cells relative to controls. Transcriptomic analysis of the cells also revealed increased expression of genes involved in both canonical and non-canonical NF-kB pathways. These results affirmed that cytokines from ACM-associated CMs may play an overlooked yet important role in recruiting immune cells to the diseased heart.

### 5.3. DSP Mutant hiPSC Lines 

Currently, only three studies have been performed with hiPSCs containing *DSP* mutations. Ng et al. reported a large single-family dACM cohort harboring a *DSP* variant (p.R451G) [91]. Immunostaining of the patient’s autopsy sample showed reduced DSP and Cx43 signals in the ICD compared with a donor control. Examination of an hiPSC line with the same *DSP* mutation (R451G) generated from a related family member established that this mutation leads to lower protein levels of Cx43, although a higher percentage of phosphorylated Cx43, relative to controls. The authors suggested that increased ubiquitin-mediated degradation of gap junctions may account, in part, for the reduced Cx43 levels. They also found that conduction velocity was similar between *DSP* mutant engineered heart tissues (EHTs) and controls. To understand the mechanism behind reduced DSP, the authors used in silico modeling to show that the missense DSP mutation could increase the mutant protein’s vulnerability to calpain-mediated degradation. Subsequent application of a calpain inhibitor to the homozygous mutant hiPSC-CM EHTs led to an increase in DSP protein abundance. Since normal hiPSC-CM EHTs exposed to the same calpain inhibitor had a decreased level of DSP, the effect of this inhibitor was conditional on the genotype [91]. *DSP* mutant EHTs also exhibited increased time-to-peak force with comparable peak amplitudes and relaxation kinetics compared with controls [91]. In contrast, in another EHT system [93] these contractile defects were not apparent in hiPSC-CMs harboring a compound heterozygous DSP mutation (c.273+5G>A and p.R2229X2261). Only when the cells were subjected to mechanical load could contractile differences be observed. Specifically, when cultured as dynamic EHTs (dyn-EHTs), these mutant DSP hiPSC-CMs showed significant contractile shortening and increased diastolic stress relative to controls [93]. Finally, Gusev et al. investigated one other DSP variant (p.H1684R) [92]. Although immunofluorescence revealed no changes in DSP or PKP2 protein localization at cell junctions, the authors found reduced *I*_Na_, *I*_CaL_, and *I*_KATP_ along with increased *I*_to_ relative to control cells. These alterations were observed alongside reduced triggering probability, amplitude, and duration of APs [92]. Altogether, CMs differentiated from these DSP mutant lines retained cell phenotypes consistent with proarrhythmic events, and tissue engineering strategies proved to be invaluable in generating physiological models capable of recapitulating dACM phenotypes.

### 5.4. DSC2 Mutant hiPSC Lines

To the best of our knowledge, only one *DSC2* mutation has been studied in hiPSC-CMs. In a triad of studies published within the past few years, the Chevalier group characterized the electrical, calcium, and contractile phenotypes of hiPSC-CMs generated from a patient with a DSC2 variant (c.394C>T). Relative to controls, mutant cells had shorter APDs, which was accompanied by a decreased density of voltage-gated Ca^2+^ and increased density of voltage-gated K^+^ currents. [88,185]. RNA encoding the potassium channels hERG and K_v_7.1 also increased [186]. An analysis of consecutive APs revealed greater spontaneous AP firing rate in mutant relative to control cells [88] as well as a greater percentage of cells presenting with aberrant electrical events (EAD- or DAD-like episodes) [186]. Compared with control cells, mutant cells had reduced calcium amplitude and decay time, increased frequency of spontaneous calcium transients, and greater propensity for aberrant calcium-spark-induced events, the latter of which is indicative of ryanodine receptor leakage [88,186]. These effects were further reflected in aberrant contractile patterns in the cells: increased contraction rate, reduced duration of contraction, and greater asynchrony of contraction across the monolayer [88]. Remarkably, exposure of the mutant cells to either a PPARƔ inverse agonist [185] or a mineralocorticoid receptor antagonist [186] attenuated many of these electrical, calcium, and contractile anomalies. Through these studies, several therapeutic candidates were identified that ameliorate the observed electrophysiological remodeling within ACM-CMs.

### 5.5. Desmin Mutant hiPSC Lines

A small number of hiPSC lines have been created from patients with desmin-related CMPs (a subtype of desminopathies). The line generated by Protonotarios et al. from a patient with a malignant form of biventricular ACM (*DES*, p.L115I) led to disruption of the desmin filament with cytoplasmic aggregation of mutant DES [187]. Brodehl et al. utilized normal hiPSCs coupled with an expression plasmid that encoded a *DES* mutation (c.364T>C, p.Y122H) to show that this mutation was pathogenic and likely a contributing factor to the development of a restrictive CMP in a patient [188]. Khudiakov et al. also generated one line carrying a splice site mutation (c.735+1G>A) in the *DES* gene. Although deleterious DES mutations would be expected to show abnormal desmin network formation with impaired desmin–protein interactions, these authors did not perform any functional analyses, making it difficult to assess the consequences of this mutation on disease development [189].

## 6. Conclusions and Future Prospects

Arrhythmogenic cardiomyopathy is a familial disease that in a subset of patients can be linked directly to mutations in desmosomal genes. Genetic variants in *PKP2, DSG2, DSP, JUP,* and *DSC2* can lead to dysfunctional cell-to-cell adhesions and inadequate development of the ICD. This may result in the mechanical failure of the ICD to effectively tether/bind adjacent CMs, a failure of which may result in intercellular gaps. Loss of cell-to-cell structural integrity, gap junction connectivity, and appropriate SCN5A localization likely contributes to various pro-arrhythmic mechanisms. These changes may also be related to changes in cell signaling that in the long term induces fibrofatty myocardial replacement. Most of these results have been identified using human genetic studies coupled with an analysis of human tissue samples and primary human cells. However, human cardiac disorders are often complex, polygenic conditions that are strongly influenced by environmental and genetic factors [190]. In the case of dACM, this is confounded by the relatively low penetrance observed among some carriers of desmosomal gene mutations.

To better define mechanisms associated with dACM, rodent models have been developed and extensively evaluated. These studies have confirmed the role of desmosome gene mutations in the manifestation of dACM. Morphological and electrophysiological attributes of dACM have been recapitulated, and some possible interventions to ameliorate disease progression have been identified. Problematically, human and animal model systems are quite complex and, electrophysiologically, quite different. Inflammatory responses as potentially critical components of dACM progression and pathology have also been observed. The unraveling of specific disease mechanisms using animal cells, thus, is complicated by differences across animal species and by variations among cardiac adaptive processes that can occur at cell-, tissue-, organ-, inflammatory-, and organism-based levels. Despite these limitations and the cost of animal studies, the availability of cells and tissues and the ability of researchers to monitor (invasively and non-invasively) disease progression in animals have led to important mechanistic insights into dACM.

Human iPSC-derived CMs represent a reproducible model system for basic cell biology and genetic research, which readily can be applied to the study of specific gene mutations. Over the past 10 years, many research groups have contributed to our understanding of dACM through hiPSC-CM models of dACM (see Section 5). The resulting CMs, cultivated either in 2D or 3D, have recapitulated many key phenotypes, including some that result in structural (ICD, desmosomes, and sarcomeres) and functional (ion channel currents and disruption of intercellular connections) changes found to be associated with dACM in patients. Using hiPSC models, GSK3β and Wnt/β-catenin signaling pathways have been confirmed to be critical early components of the disease process. Dysregulation of a mechanosensory pathway and MRTF/SRF activation was found to underlie the formation of fibrofatty structures in myocardium, at least in vitro. The presence and activation of a cell-based immune response involving NFkB that may contribute to disease progression has also been identified. We have assembled the key findings from these studies into tables, broadly organized according to molecular profiling of the desmosome and ICD proteins (Table 1), electrophysiology (Table 2), and calcium and contraction (Table 3). Our aim in compiling these results is to facilitate comparison of major molecular- and cellular-level phenotypes across studies where different lines and desmosomal mutations have been analyzed. We invite the reader to peruse this information to gain a better understanding of the ACM-related work that has been performed in hiPSC-CMs. 

Although the reproducibility of generating hiPSC-CMs bodes well for the systemic in vitro analysis of dACM, a number of hiPSC-associated limitations should be considered. First, most published studies with hiPSC-CMs do not have isogenic control lines. Genome-wide profiling has shown that 5–46% of the hiPSC phenotype variability among lines, including differentiation capacity and cellular morphology, results from differences among individuals and from polygenic variability [191]. Such a potentially high degree of phenotypic variability requires that researchers perform experiments on CMs derived from multiple hiPSC lines as well as through the use of appropriate isogenic control lines (e.g., CRISPR/Cas9 edited lines). Otherwise, data misinterpretation is a potential concern. Furthermore, once isogenic lines are created, in vitro studies can be designed to account for both genetic as well as environmental or ancillary factors that may contribute to the manifestation of disease processes. Second, in vitro differentiated hiPSC cultures contain both non-myocytes and phenotypically different myocytes (atrial-, ventricular-, and pacemaker-CMs). New and improved protocols to eliminate non-myocytes have been reported that require the use of cell sorting and markers like SIRPA or LSMEM2 [192] or CM enrichment with lactate-mediated metabolic selection [193]. Third, hiPSC-CMs generally experience a “maturation arrest” even after prolonged culture [194], which raises questions about their relevance and translatability to dACM phenotypes manifested in adult CMs. Some aspects of cell maturation can be resolved through cell sorting with markers like CD36, use of small molecules or fatty acids, or use of bioengineered tissues [195,196,197]. Current strategies to drive hiPSC-CMs to a structural and functional state akin to adult CMs are summarized elsewhere [198,199,200,201,202] and represent a significant area of investigative research in the community of developmental cardiology. Fourth, and depending on the affected gene, dACM can preferentially affect heart function in a chamber-specific manner. Recent advances in developmental biology coupled with improvements in hiPSC differentiation protocols, however, have begun to allow for the isolation of chamber-specific ventricular CMs derived from the first (left ventricle) and second (right ventricle) heart field through differentiation in the hiPSC culture system using protocols developed by Yang et al. and others [30]. 

Despite potential limitations to the use of hiPSC-CMs to study dACM, a number of important research questions are amenable to investigation using these model systems. First, the immaturity of hiPSC-CMs may be apt for developmental studies designed to study ICD formation during the perinatal period. Postnatally, the desmosome proteins are distributed initially in a punctate pattern at or near the cell membrane [35], and the mechanical junctions do not show an adult-like pattern until after one year of age [56]. Since hiPSC-CMs are fetal-like and desmosome proteins also show a punctate pattern in 2D cultures (Figure 2), this model system should be apt for understanding how the ICD forms during the perinatal period. Through the use of engineered heart tissues or patterned substrates, it should be possible to develop a model where the AJs form, followed by the assembly of the desmosomes, and the subsequent colocalization of ion channels and gap junctions to the ICD [44] (see Figure 2). Using gene editing and molecular or pharmacological approaches, individual proteins or signaling mechanisms could then be targeted to unravel the process of ICD formation and how mutations in desmosomes adversely affect the development of aligned ICD-like structures at the perimeters of CMs.

Second, genetic testing has revealed the presence of unclassified genetic variants with ambiguous pathogenicity known as variant(s) of uncertain significance (VUS). Since the penetrance of some dACM-associated mutations is low, or the phenotype may be concealed during early phases of the disease, the early identification of pathogenic variants among the VUS may be critical to patient treatments. Ma et al. demonstrated that hiPSC-CMs can be employed to determine the pathogenicity of genomic VUS when coupled with CRISPR/Cas9 technologies [172]. Using genome-edited carrier-specific hiPSC-CMs, they classified both benign and pathogenic phenotypes in a carrier-specific manner for mutations linked to hypertropic cardiomyopathy. More specifically, they reproduced the asymptomatic phenotype in hiPSC-CMs of a likely pathogenic variant in the MYL3 gene (*MYL3*_(170C>A)_), confirmed a benign genome-edited isogenic homozygous VUS *MYL3*_(170C>A)_ line and heterozygous frameshift mutation *MYL3*^(170C>A/fs)^ line associated with an asymptomatic clinical phenotype, and validated the ability of iPSC-CMs coupled with genome-editing to recapitulate a pathogenic phenotype of a symptomatic carrier. These proof-of-principle results illustrate how hiPSC platforms, including those designed to study dACM, can be employed to assess VUS for pathogenicity and to perform personalized risk-assessments of patient-specific VUS.

Third, inflammation may be a major contributor to the disease process of dACM [45] and patients may experience repeated episodes of myocarditis [203,204]. An understanding of the inflammatory response could provide valuable information on how to treat dACM in patients. While hiPSC-CMs may be useful for in vivo analyses of inflammation, problematically, the introduction of these cells into animal models can also cause inflammatory cascades. Alternatively, the finding that hiPSC-CMs express elevated cytokines/chemokine levels in vitro, which may be mediated through NFkB-mediated inflammatory pathways, is consistent with the hypothesis that an innate cell immune response characterized by cytokines contributes to or is responsible for the pathological phenotype (altered electrophysiology) of DSG2-mutant CMs in vitro. The question is whether other mutations in DSG2 and other desmosomal genes lead to the activation of the same innate immune response and whether the responses are affected by the genetic backgrounds. Moreover, it will be valuable to determine to what extent the degree of cell maturation matters in terms of distinguishing dACM mutant cells from normal cells. To address these issues, researchers need to determine whether innate inflammatory signals associated with the disease phenotype are enhanced in dACM hiPSC-CMs with increased maturation, and whether inhibition of the innate inflammatory response in hiPSC-CMs with desmosomal mutations can prevent or improve the electrophysiological and calcium parameters of these cells.

Fourth, disease-specific hiPSCs are very useful tools for study of the pathophysiology and cellular responses to therapeutic agents. As described earlier in this review, hiPSC-CM model systems have been employed successfully to test for drug interventions (e.g., SB216763) that may limit or reverse disease processes through regulation of Wnt/β-catenin signaling. The development of more robust and reproducible hiPSC-CM systems, containing lineage-specific and maturation-defined CMs, will likely advance and accelerate the discovery of new drugs or small molecules that have the potential of improving dACM patient outcomes. Furthermore, the importance of the non-myocyte population in the myocardium, the composition of the extracellular matrix, and the degree of tissue heterogeneity in disease manifestation can be investigated using tissue engineering approaches [205,206]. However, new drugs going to market in the U.S. require Food and Drug Administration approval, which can be very costly, particularly if any of the drugs lead to arrhythmias. The most common and severe form of drug-induced arrhythmia is Torsade de Pointes (TdP), a polymorphic, life-threatening ventricular tachycardia which can result from re-entrant electrical activity or multiple ectopic sites. TdP has been associated with prolongation of the QT interval in the electrocardiogram, and blockage of the HERG channel (I_Kr_) has been linked to TdP-associated drugs [206]. To decrease costs and to have human screens for drug development, the FDA is developing new guidelines (CiPA initiative) based, in part, on hiPSC-CMs [205,207]. In vitro tissue models of cardiac arrhythmia like TdP are yet in their infancy, and will require the development of appropriate 2D and 3D tissue models with appropriate physiological sensors to study the underlying mechanisms of arrhythmia [205]. The use of these cells from normal and diseased lines should foster the development of safe and effective drugs that may be effective in treating patients with dACM.

Finally, the identification of molecular markers of dACM may prove invaluable to clinicians who evaluate patients, particularly if informative markers can be found among dACM patients with different affected genes or gene mutations. The use of hiPSC-CMs coupled with transcriptomic (RNA) or proteomic (whole protein, surfaceome, secreted proteome, and glycosylation) assays should prove invaluable for this purpose, particularly since the cells can be reproducibly generated and assayed. RNA-seq, for example, provides an unbiased approach to evaluate transcripts and, by extension, potential markers that may be elevated or dysregulated in dACM. Proteomic technologies, by contrast, can specifically identify and investigate whole proteome and subproteomes, as well as the secretomes of hiPSC-CMs, in a quantitative and discovery-driven manner [208]. Once a protein or protein panel has been identified via mass spectrometry, the marker(s) should permit the identification and tracking of affected cells, as well as lead to the development of assays that may distinguish among the genes and gene mutations that contribute to dACM. Moreover, the presence of a glycan moiety within a polypeptide is critical for proper protein folding and stability. Changes in glycosylation can be assessed using proteomic approaches [208] and, in the cases of DSC2 and DSG2, molecular studies have been published suggesting that improper glycosylation can lead to inappropriate cell localization [209,210]. This is a relatively poorly studied component of dACM, and investigations into altered glycosylation may prove critical to understanding disease manifestation in patients.

In conclusion, hiPSC-derived CMs represent a reproducible model system for basic cardiac cell biology, pharmacology, and translational research applicable to dACM. In the future, the use of heart-field-specific and developmentally staged (matured) hiPSC-CMs has the potential of increasing our understanding of the mechanisms responsible for the development and progression of dACM, as well as dissecting why patients may have chamber-restricted phenotypes or are not affected by gene mutations in desmosomal proteins. Given the research advances described in this review over the past few years, we predict that well-defined dACM hPSC-CMs will play an increasingly important and informative role for furthering research into gene-caused disease processes that will ultimately lead to the development of new translational medical approaches capable of treating patients with dACM.

## Figures and Tables

**Figure 2 genes-14-01864-f002:**
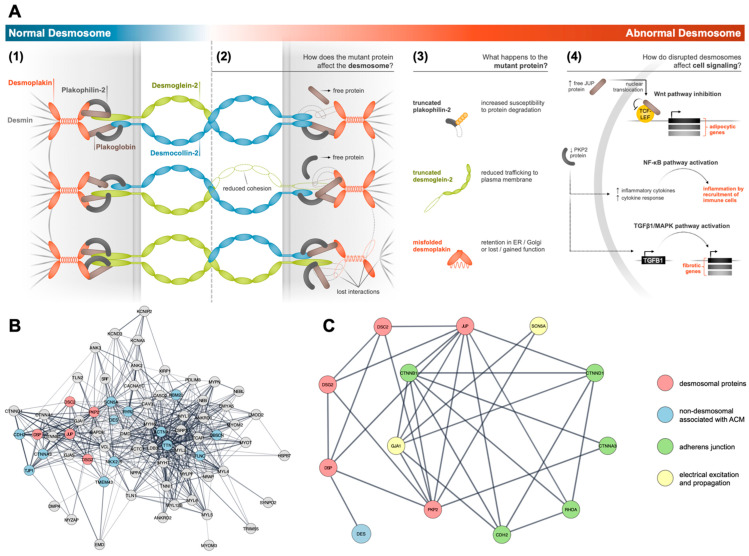
Architecture of cardiac desmosomes and interactions within the ICD. (**A**) Schematic of normal and abnormal desmosome architecture, where the abnormal condition reflects reported pathogenic phenotypes in ACM. In a normal desmosome (1), the desmosomal cadherins, desmocollin-2 and desmoglein-2, interact extracellularly to mechanically connect adjacent cells. Intracellularly, plakoglobin and plakophilin-2 serve as adaptor proteins to link the cadherins to desmoplakin. Desmoplakin links the desmosome to the desmin intermediate filament network within the cell. In an abnormal desmosome (2), mutant proteins (3) compromise desmosome structure while causing several other ancillary effects within the cell, including dysregulated cell signaling (4). Hyperactive or inhibitory signaling through plakoglobin or plakophilin-2 can result in a range of ACM-related phenotypes, such as adipogenesis through Wnt inhibition [51], fibrosis through TGFB1 upregulation [52], or recruitment of inflammatory cells to the heart through NF-KB [53]. (**B**) Graph of interactions between relevant components of the ICD, which include desmosomal proteins (red), non-desmosomal proteins (blue) that have been associated with ACM. Interaction data sourced from STRING database and cross-validated with NCBI gene repositories and various review papers [54,55]. (**C**) Selected interactive data showing the interconnected links among the desmosomal proteins, AJs, and non-desmosome proteins associated with ACM.

**Figure 3 genes-14-01864-f003:**
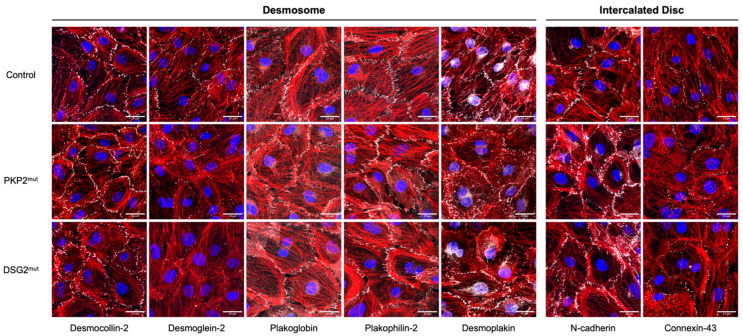
Immunostaining of proteins of the desmosome and cardiac ICD in ACM hiPSC-CMs. Pluripotent stem cells from (1) a donor with no known mutations associated with ACM (top row, control), (2) a patient with ACM and a heterozygous PKP2 truncating mutation (middle row, PKP2^mut^) [58], and (3) a patient with ACM and a heterozygous DSG2 truncating mutation (bottom row, DSG2^mut^) [59] were differentiated into CMs, which were labeled for markers of the desmosome (DSC2, DSG2, PKG, PKP2, and DSP) and intercalated disc (N-cad, Cx43). In normal hiPSC-CMs, these proteins are expressed in a characteristic punctate pattern along the length of the membrane where neighboring myocytes are mechanically connected at cell junctions, yet disease-associated hiPSC-CMs display variations in the intensity and/or patterning from the control cells. The monolayer preparations of hiPSC-CMs shown were imaged at day 35 of in vitro CM differentiation. All cells were counterstained with DAPI and phalloidin (actin) to demarcate individual cells. Scale bars: 25 μM.

**Figure 4 genes-14-01864-f004:**
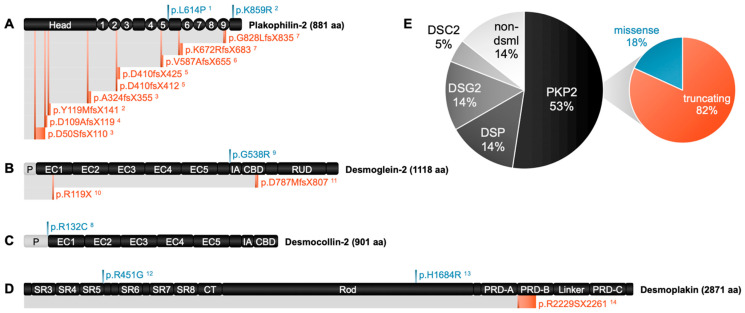
dACM-associated mutations studied in hiPSC-CMs to date. Primary structure schematics [82] for (**A**) plakophilin-2, (**B**) desmoglein-2, (**C**) desmocollin-2, and (**D**) desmoplakin are shown with all known variants studied in hiPSC-CMs mapped to their respective protein in p-dot (p.) nomenclature (superscripts denote references to primary source—see end of caption). Note that the “P” domain in light gray at the N-terminus of desmocollin-2 and desmoglein-2 represents amino acids which are part of the preproprotein and are eventually cleaved off to yield a mature protein. Missense variants are denoted in blue, and truncating variants are denoted in red. In this diagram, a line branching from a protein indicates the site where a specific mutation causes an amino acid switch, and a red band running parallel to the protein of interest from the switch site represents how far downstream the protein is translated before a premature stop codon is reached. (**E**) Proportion of ACM studies in hiPSC-CMs attributable to mutations of specific genes: *PKP2* (plakophilin-2), *DSP* (desmoplakin), *DSG2* (desmoglein-2), *DSC2* (desmocollin-2), and non-desmosomal genes. To the best of our knowledge, mutations in *JUP* (plakoglobin) have not yet been studied in hiPSC-CMs. In contrast, most ACM reports in hiPSC-CMs investigate *PKP2* variants, a majority of which are truncating variants over missense variants. Superscripts for references: 1 [83]; 2 [84]; 3 [58]; 4 [85]; 5 [86]; 6 [87]; 7 [8]; 8 [88]; 9 [89]; 10 [90]; 11 [59]; 12 [91]; 13 [92]; and 14 [93].

**Table 1 genes-14-01864-t001:** Molecular characterization of desmosomal and ICD proteins in hiPSC-CM models of dACM. For each mutation studied in hiPSC-CMs for which data are available, we summarize the findings related to desmosomal and ICD protein and mRNA expression in this table. Mutations are given in p-dot (p.) nomenclature and specified as T (truncating) or M (missense). Symbols and colors are described below the information presented in the table, and comparisons are represented as mutant cell lines relative to control cell lines under the same conditions as presented by the data published in the primary sources, for which the references are provided. Pkp2 = plakophilin-2; Pkg = plakoglobin; Dsg2 = desmoglein-2; Dsp = desmoplakin; Dsc2 = desmocollin-2; N-cad = N-cadherin; and Cx43 = connexin-43.

Gene	Mutation	T/M	Pkp2	Pkg	Dsg2	Dsp	Dsc2	N-cad	Cx43	Na_v_1.5	Reference
PKP2	p.L614P	M	▼▽	▼▽		■□		■□	■□		[173]
	p.G828LfsX835	T	▼	▼						▽	[28]
	p.K672RfsX683	T	▼▽	▼							
	p.A324fsX355	T	▼▽	▼□		□			▼		[58]
	p.D50SfsX110	T	▼▽	▼□		□			▼		
	p.V587AfsX655	T	▼▼□			▼					[87]
	p.A324fsX355	T	▽	▼▼▽		▽		▽	▼▽	▼▽	[179]
	p.G828LfsX835	T	▼□	▼□					▽	▼▽	[178]
	p.Y119MfsX141 + p.K859R	T+M	▼ ▼	▼ ■						■■□	[84]
	p.D109AfsX119 *	T	▼▼▽			▼		■	▼		[85]
	p.D109AfsX119 *	T	▼▼▽			▼		■	▼		
	p.D410fsX425	T	▼▼▽	■ ■	■ ■	■	■ ■	■ ■	■ ■		[86]
	p.D410fsX412 *	T	▼▼▽	▼ ▼	▼ ▼	▼	▼ ▼	▼ ▼	■ ■		
DSG2	p.G638R^†^	M			NA					▽	[89]
	p.D787MfsX807	T	■■▽	■■□	▼▼▽				□	△	[59]
	p.R119X	M	■ ■	■ ■	▼▼▽	■ ■		■	■		[90]
DSP	p.R451G	M				▼			▼		[91]
	p.H1684R	M	■			■ ■					[92]
	c.273+5G>A + p.R2229X2261	T				▼▼▽	▼				[93]
DSC2	p.R132C	M		▽			▼▽				[88]
Heterozygous mutation (not shaded)						▼ decreased relative to control					
Compound heterozygous mutation (shaded red)						■ no change relative to control					
Homozygous mutation (shaded blue)						▲ increased relative to control					
Mutant gene product (shaded gray)						▲■▼ membrane protein (immunofluoresence)					
* Artificially introduced mutation (not patient derived)						▲■▼ total protein abundance (Western blot)					
† Assumed heterozygous, although not specified						△□▽ total mRNA abundance (qPCR / RNA-seq)					

**Table 2 genes-14-01864-t002:** Electrophysiology in hiPSC-CM models of dACM. For each mutation studied in hiPSC-CMs for which data are available, we summarized findings related to cardiomyocyte electrical activity, including molecular and functional data for individual ion channels and parameters related to excitation, conduction, and synchrony in multicellular preparations. Mutations are given in p-dot (p.) nomenclature and specified as T (truncating) or M (missense). Symbols and colors are described below the information presented in the table and comparisons are represented as mutant cell lines relative to control cell lines under the same conditions as presented by the data published in the primary sources, for which references are provided.

									Molecular Data									Single Channel Currents										Excitation, Conduction, & Synchrony		
Gene	Mutation	SCN5A/Nav1.5	GJA/Cx43	SK3/KCa2.3	SK4/KCa3.1	KCNJ11/Kir6.2	KCNQ1/Kv7.1	KCNJ2/Kir2.1	KCNH2/Kv11.1	SLC8A1/NCX1	INa	ICaL	INCX	Ito	IK	Ikr	Iks	ISK	ISK4	IKATP	AP probability	Resting potential	AP upstroke velocity	AP upstroke heterogeneity	AP amplitude	AP duration	AP frequency	Conduction velocity	Arrhythmic events	Refs.
PKP2	p.G828LfsX835	▽									▼											■	▼		▼	■				[28,68]
	p.A324fsX355		▼																				▼							[58]
	p.Y119MfsX141 + p.K859R	■■□									▼												▼			■				[84]
	p.D109AfsX119 *		▼																							▲				[85]
	p.D109AfsX119 *		▼																							▲				
DSG2	p.G638R ^†^	▽		▽	▲△	▽					▼	■	▼	▼		▲	■	▼	▲	▼		■	▼		▼	■	▲		▲	[89,182]
	p.D787MfsX807	△	□				△	□		□													■	▲		▼		■		[59]
	p.R119X																											▼	▲	[90]
DSP	p.R451G		▼																									■		[91]
	p.H1684R										▼	▼		▲							▼				▼	▼				[92]
DSC2	p.R132C						△		△		▼				▲							■	■		■	▼	▲		▲	[88,185,186]
Heterozygous mutation (not shaded)											▼ decreased relative to control																			
Compound heterozygous mutation (shaded red)											■ no change relative to control																			
Homozygous mutation (shaded blue)											▲ increased relative to control																			
											▲■▼ membrane protein (immunofluoresence)																			
* Artificially introduced mutation (not patient derived)											▲■▼ total protein abundance (Western blot)																			
† Assumed heterozygous, although not specified											△□▽ total mRNA abundance (qPCR/RNA-seq)																			

**Table 3 genes-14-01864-t003:** Calcium, force, and contraction in hiPSC-CM models of dACM. For each mutation studied in hiPSC-CMs for which data are available, we summarized findings related to cardiomyocyte calcium handling, force generation, and contraction. Mutations are given in p-dot (p.) nomenclature and specified as T (truncating) or M (missense). Symbols and colors are described below the information presented in the table and comparisons are represented as mutant cell lines relative to control cell lines under the same conditions as presented by the data published in the primary sources, for which references are provided.

								Molecular Data					Calcium								Force, Stress, & Contraction						
Gene	Mutation	RYR2/RyR2	CACNA1C/Cav1.5	SLC8A1/NCX1	ATP2A2/SERCA2a	Diastolic intra Ca2+	Systolic intra Ca2+	Intra Ca2+ AUC	Intra Ca2+ time-to-peak	Intra Ca2+ decay time	Ca2+ transient frequency	Ca2+ sparks	EAD or DAD events	Peak force	Time-to-peak force	Contraction rate	Tissue diastolic length	Tissue shortening	Diastolic stress	Systolic stress	Systolic	Twitch stress	Contraction velocity	Relaxation duration	Aberrant contraction	Contraction asynchrony	Refs.
PKP2	p.G828LfsX835	□	△	□	▽					■																	[28]
	p.D109AfsX119 *													▼						▼	▼						[85]
	p.D109AfsX119 *													▼						▼	▼						
	p.D410fsX425																						▼				[86]
	p.D410fsX412 *																	▼									
DSG2	p.G638R ^†^					■	■						▲														[89]
	p.D787MfsX807								▼	▼																	[59]
	p.R119X													▼													[90]
DSP	p.R451G									■				■	▲												[91]
	c.273+5G>A + p.R2229X2261																▼	▼	▲			▼					[93]
DSC2	p.R132C					▼		▼		▼	▲	▲				▲		▲					■	▼	▲	▲	[88,185,186]
Heterozygous mutation (not shaded)													▼ decreased relative to control														
Compound heterozygous mutation (shaded red)													■ no change relative to control														
Homozygous mutation (shaded blue)													▲ increased relative to control														
													▲■▼ membrane protein (immunofluoresence)														
* Artificially introduced mutation (not patient derived)													▲■▼ total protein abundance (Western blot)														
† Assumed heterozygous, although not specified													△□▽ total mRNA abundance (qPCR/RNA-seq)														

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
