# Peer review of "Understanding Arrhythmogenic Cardiomyopathy: Advances through the Use of Human Pluripotent Stem Cell Models"

_genes, 2023, doi:10.3390/genes14101864_

Round 1

Reviewer 1 Report

In the review article 'Understanding Arrhythmogenic Cardiomyopathy: Advances through the Use of Human Pluripotent Stem Cell Models, submitted by Christianne J. Chua and coworkers to Genes, the authors summarize the knowledge about ACM with a focus on iPSC-derived cardiomyocytes. The topic of this review is highly relevant.

However, I would change some minor points:

1.) Figure 1: Could you add for each ACM gene a reference in the figure legend? This figure is pretty good, but it could even better, if the reader can find references for each desmosomal and non-desmosomal gene. 

2.) In Figure 1: DSC2 not DSC and DSP and not DSP2.

3.) In Figure 3 there is a graphical defect. Could you please repair this?

4.) Paragraph 4.3 Could you add a reference for the different splicing forms?  I think there are only two different forms? 

5.) Could you also discuss the relevance of sugar residues in DSC2 and DSG2? Previously there are reprots, which have investigated the post-translational modificiactions in desmocollin-2 and desmoglein-2.

6.) I would also add a paragraph about desmin (DES), since desmin filaments are directly coupled to the desmosomes and several DES mutations cause ACM. Therefore, it makes sense to include desmin, although it is not a direct desmosomal protein. 

7.) Recently, the relevance of mutations in the prodomain of desmocollin-2 was shown for ACM. Could you include and discuss this point in the revised version of your manuscript.

However, in summary, the authors have written a nice review article, which should be published after a major revision. I am optimistic that the authors can improve the quality at specific points.

The manuscript should be double-checked by a native-speaking editor. 

Author Response

We would like to thank the reviewer for his/her comments on our review article for Genes. Your suggestions were very helpful, and we believe that with this revision, the Review is greatly improved, and we hope that it will be acceptable for publication.

  • Figure 1: Could you add for each ACM gene a reference in the figure legend? This figure is pretty good, but it could even better, if the reader can find references for each desmosomal and non-desmosomal gene. 

We have added references for the desmosomal and non-desmosomal genes in the Figure Legend as suggested. Furthermore, we removed several panels/boxed that were related to Cardiomyopathies, but which were not relevant to this review article. The figure has therefore been revised, and we think that the information is more apt for the reading audience interested in this topic.

  • In Figure 1: DSC2 not DSC and DSP and not DSP2.

Thank you for noting this mistake. We have of course corrected the gene names in this figure to make the information more accurate and accessible.

  • In Figure 3 there is a graphical defect. Could you please repair this?

This has been corrected in the revision. Thank you for noting this defect.

  • Paragraph 4.3 Could you add a reference for the different splicing forms?  I think there are only two different forms? 

Thank you for this comment. I have corrected this mistake, as I had meant to refer to 4 transcripts, two of which exist due to different initiation sites of transcription, as well as the two splice variants. The text has been corrected to address this issue.

  • Could you also discuss the relevance of sugar residues in DSC2 and DSG2? Previously there are reprots, which have investigated the post-translational modificiactions in desmocollin-2 and desmoglein-2.

Thank you for bringing this to our attention. In the revision, we have briefly discussed the relevance of glycan residues in DSC2 and DSG2 in terms of post-translational modifications. When we reviewed the literature, there was good evidence for DSC2 but the data regarding DSG2 was, in our opinion, less concrete. We do however report this information in Section 6 and provide references in support of this data; however, we have also suggested that this is likely an important topic for future investigations. As we have some publications in this area with normal in vitro differentiated cardiomyocytes, we have also briefly discussed how changes in glycosylation and its relevance to ACM might be addressed.

  • I would also add a paragraph about desmin (DES), since desmin filaments are directly coupled to the desmosomes and several DES mutations cause ACM. Therefore, it makes sense to include desmin, although it is not a direct desmosomal protein. 

I have added two paragraphs on DES in this paper. The first addresses desminopathies in human and animal models in a new Section 4.6 and a second discusses DES in hiPSCs in a new Section 5.5. However in terms of hiPSC, desmin and ACM, the data available for discussion is very limited.

  • Recently, the relevance of mutations in the prodomain of desmocollin-2 was shown for ACM. Could you include and discuss this point in the revised version of your manuscript.

In section 4.3, we have added a paragraph that directly addresses this point. Thank you for bringing this to our attention.

The manuscript should be double-checked by a native-speaking editor. In revising this review article, we did find a few grammatical errors, which we have corrected. As all the authors are native English speakers, we regret this earlier oversight, but we believe that the article is grammatically correct.

Reviewer 2 Report

In this review, the authors provide a summary of hiPSC-CM models in Arrhythmogenic cardiomyopathies. I have read it with great interest and found it informative. This is an area with increasing interest especially in the emergence of gene therapeutics and other mechanism-based drug targets. My comments follow:

- A major point I want to raise is that of nomenclature. The term ACM is not a widely accepted one. It encompasses a very wide range of diseases that have little in common on an aetiological and mechanistic basis. This is reflected in the just published 2023 ESC guidelines for the management of cardiomyopathies, where the focus is moved from labelling diseases from their phenotypes to their aetiological basis. Since this review focuses on the desmosomal genes I would suggest that you refer to either ARVC phenotype or desmosomal cardiomyopathies. This will give greater clarity to the readers for the years to come.

- Similarly, in regard to the classification presented in Figure 1, these are all phenotypes. Ischaemic heart disease is not really a cardiomyopathy and this has been removed from the definitions of cardiomyopathy for many years now. Figure 1 is not particularly useful for the purpose of this review.

- In regards to the introduction, there are elements that make it rather unfocused. It can be shortened to reflect more about the problems that ARVC patients phase and how hipsc technology can unlock better therapies or diagnostics. 

- I would find it much more stimulating if the future prospects chapter more expanded and better structured. In particular how can we use those models to understand the pathogenicity of VUSs (a very common problem in practice). Is there a molecular signature of disease that can be utilized? Many of our patients experience multiple myocarditis episodes. How can inflammation be modelled and studied in these models? Finally, how can these models contribute to the huge effort currently undertaken to intrioduce gene therapy in these patients?

Author Response

(The authors gave the same response as above.)

Reviewer 3 Report

The authors have devoted an extensive amount of effort in elucidating the current progress of arrhythmic cardiomyopathy, with a focus on those state-of-the-art research progresses highlighted by iPSCs. This is important in enlightening readers about future studies that can leverage iPSCs to unravel further molecular mechanisms of arrhythmic cardiomyopathy. I think the review entails sufficient in-depth details and at the same time maintains a nice flow for readers to follow. The following minor comments might be addressed by the authors to improve it.

1.       Page 3, line 2, please indicate clearly what is the 112% in ratio of. Probably the normal value predicted by the age and surface area. Please clarify. Also, the authors have used a reference which the value of 112% was established beginning of the century. Please check the updated value.

2.       In Figure 2A, it may be easier for readers to understand the correlation of abnormal desmosomes with arrhythmic cardiomyopathy if another panel is added on the right that connects the desmosome changes with apoptosis, adipogenesis, and fibrosis respectively that corresponds with each of the theory proposed. In Figure 2B, it is unknow that the authors mean by interaction of the ion channels and desmosomes. Is it structural and physical interaction or other types of action? The meaning of the lines that connect the different points are also not explained.

3.       Figure 3: grey and white texts are really hard to see.

Author Response

We would like to thank the reviewer for his/her comments on our review article for Genes. Your suggestions were very helpful, and we believe that with this revision, the Review is greatly improved, and we hope that it will be acceptable for publication.

  1. Page 3, line 2, please indicate clearly what is the 112% in ratio of. Probably the normal value predicted by the age and surface area. Please clarify. Also, the authors have used a reference which the value of 112% was established beginning of the century. Please check the updated value.

Thank you for this comment. We were able to update the value; however, due to comments from Reviewer #2, we needed to condense our introduction and make it more relevant for ACM. Ultimately, we removed this section from the Revised manuscript.

  1. In Figure 2A, it may be easier for readers to understand the correlation of abnormal desmosomes with arrhythmic cardiomyopathy if another panel is added on the right that connects the desmosome changes with apoptosis, adipogenesis, and fibrosis respectively that corresponds with each of the theory proposed. In Figure 2B, it is unknow that the authors mean by interaction of the ion channels and desmosomes. Is it structural and physical interaction or other types of action? The meaning of the lines that connect the different points are also not explained.

Thank you for this comment, and we agree (upon review of the figure) that this could have been improved. We have modified Figure 2 in several ways, and in fact, it is largely a new figure. We have also included a panel to the right of the prior (but now modified schematic) where we link desmosomal changes with transcriptional and other adaptive events that lead to specific phenotypes. Moreover, Figure 2B and newly added 2C have been completely modified to better describe what we mean by ‘interactions”. We have also explained this latter point directly in the text, which had not been done before.

  1. Figure 3: grey and white texts are really hard to see.

Thank you for bringing this to our attention. We think that the grey and white texts were a result of the uploaded figure, which appears to be low-resolution for review. We have tried to improve the resolution of all the figures in this resubmission, and we will try to confirm that any submitted documents are of high enough resolution to clearly understand the contents of the illustrations that we provide. Regardless, we will ensure that the journal has a high-resolution document for any possible publication.

Round 2

Reviewer 1 Report

Congratulations! The authors have significantly improved their manuscript. I suggest to accept this manuscript for publication.

Reviewer 2 Report

I am happy to accept this manuscript without any further changes.